# MftG is crucial for ethanol metabolism of mycobacteria by linking mycofactocin oxidation to respiration

Ana Patrícia Graça[1], Vadim Nikitushkin[1,2], Mark Ellerhorst[1,2], Cláudia Vilhena[3], Tilman E Klassert[4,5], Andreas Starick[6,7], Malte Siemers[6,7], Walid K Al-Jammal[8], Ivan Vilotijevic[8], Hortense Slevogt[4,5], Kai Papenfort[6,7], Gerald Lackner[1,2,7]*

[1]Leibniz Institute for Natural Product Research and Infection Biology – Hans Knöll Institute, Junior Research Group Synthetic Microbiology, Jena, Germany; [2]University of Bayreuth, Chair of Biochemistry of Microorganisms, Kulmbach, Germany; [3]Leibniz Institute for Natural Product Research and Infection Biology– Hans Knöll Institute, Department of Infection Biology, Jena, Germany; [4]Respiratory Infection Dynamics, Helmholtz Centre for Infection Research - HZI Braunschweig, Braunschweig, Germany; [5]Department of Respiratory Medicine and Infectious Diseases, Hannover Medical School, German Center for Lung Research (DZL), BREATH, Hannover, Germany; [6]Friedrich Schiller University Jena, Institute of Microbiology, Jena, Germany; [7]Microverse Cluster, Friedrich Schiller University Jena, Jena, Germany; [8]Friedrich Schiller University Jena, Institute of Organic Chemistry and Macromolecular Chemistry, Jena, Germany

*For correspondence:
gerald.lackner@uni-bayreuth.de

Competing interest: The authors declare that no competing interests exist.

## eLife Assessment

Graca et al. reports a **fundamental** missing link in the ethanol metabolism of mycobacteria and illuminates the role of a flavoprotein dehydrogenase that acts as an electron shuttle between an uncommon redox cofactor and the electron transport chain. Overall, the data presented are **compelling**, supported by a range of well designed and meticulous experiments. The findings will be of broad interest to researchers investigating bacterial metabolism.

**Abstract** Mycofactocin is a redox cofactor essential for the alcohol metabolism of mycobacteria. While the biosynthesis of mycofactocin is well established, the gene *mftG*, which encodes an oxidoreductase of the glucose-methanol-choline superfamily, remained functionally uncharacterized. Here, we show that MftG enzymes are almost exclusively found in genomes containing mycofactocin biosynthetic genes and are present in 75% of organisms harboring these genes. Gene deletion experiments in *Mycolicibacterium smegmatis* demonstrated a growth defect of the Δ*mftG* mutant on ethanol as a carbon source, accompanied by an arrest of cell division reminiscent of mild starvation. Investigation of carbon and cofactor metabolism implied a defect in mycofactocin reoxidation. Cell-free enzyme assays and respirometry using isolated cell membranes indicated that MftG acts as a mycofactocin dehydrogenase shuttling electrons toward the respiratory chain. Transcriptomics studies also indicated remodeling of redox metabolism to compensate for a shortage of redox equivalents. In conclusion, this work closes an important knowledge gap concerning the mycofactocin system and adds a new pathway to the intricate web of redox reactions governing the metabolism of mycobacteria.

## Introduction

Mycobacteria are a metabolically versatile group of microorganisms that are able to utilize various organic compounds for their carbon and energy metabolism. Especially environmental mycobacteria like *M. smegmatis* utilize for instance sugars, polyols, small organic acids, fatty acids, or sterols for their growth (*Tsukamura, 1966*). While obligate pathogens like *M. tuberculosis*, the causative agent of tuberculosis, are more limited in their menu, they still retain the ability to metabolize various nutrients available in the host organism (*Bloch and Segal, 1956*; *Zimmermann et al., 2017*). Intriguingly, alcohols like ethanol are readily consumed by many mycobacterial species, but the metabolic pathway for ethanol consumption requires genes related to the biosynthesis of the unusual redox cofactor mycofactocin (MFT; *Haft, 2011*; *Krishnamoorthy et al., 2019*). Indeed, the inactivation of the mycofactocin biosynthetic gene cluster *mftA-F* (*Figure 1A*) showed a severe impact on the growth of *M. smegmatis* and *M. marinum* cultivated on ethanol as the sole carbon source. For *M. tuberculosis,* a similar growth deficit on media containing ethanol and cholesterol was observed (*Krishnamoorthy et al., 2019*). In *M. smegmatis*, it was shown that the gene MSMEG_6242, encoding a MFT-associated alcohol dehydrogenase (termed Mno or Mdo), is strictly required for alcohol utilization. Mno/Mdo was also demonstrated to catalyze methanol metabolism in the same organism (*Dubey et al., 2018*) and involvement of mycofactocin in this process was suggested (*Dubey and Jain, 2019a*). Furthermore, a metabolomics study conducted in our lab revealed that MFT production in *M. smegmatis* was significantly increased on ethanol as carbon source in comparison to a control cultivated on glucose (*Peña-Ortiz et al., 2020a*). Recently, a homolog of Mdo was shown to be essential for MFT-dependent ethylene glycol oxidation in *Rhodococcus jostii* (*Shimizu et al., 2024*). All of this evidence supported the hypothesis that mycofactocin enables alcohol metabolism by acting as an electron acceptor of alcohol dehydrogenases (*Figure 1B*; *Haft, 2011*; *Krishnamoorthy et al., 2019*).

Although the role of MFT in alcohol metabolism is well established, further biological roles of this cofactor appear to exist. Intriguingly, the MFT gene locus seems to be de-repressed by long-chain fatty acid-CoA esters in *M. smegmatis,* which bind to the repressor MftR (*Mendauletova and Latham, 2022*). Furthermore, a comparison of the proteome of active and dormant cells revealed that *mftD* (Rv0694) was upregulated in the dormant state, which is associated with the persistence of *M. tuberculosis* in macrophages (*Nikitushkin et al., 2022*). Lastly, a study on the impact of mycofactocin inactivation (ΔmftD) on *M. tuberculosis* showed increased growth on glucose containing media as well as a decreased number of mycobacterial cells in a mouse model after the onset of hypoxia (*Krishnamoorthy et al., 2021*).

Mycofactocin is a ribosomally synthesized and post-translationally modified peptide (RiPP; *Ayikpoe et al., 2019*). After translation, the precursor peptide MftA is bound by its chaperone MftB and undergoes several modifications until it matures into a redox-active molecule. More precisely, its C-terminal core peptide (Val-Tyr) is decarboxylated and cyclized by the radical *S*-adenosyl methionine (rSAM) maturase MftC (*Khaliullin et al., 2017*). The cyclized core peptide (AHDP) is liberated from the precursor by the peptidase MftE (*Ayikpoe et al., 2018*). The flavoenzyme MftD further deaminates AHDP to premycofactocin (PMFT) thus creating a redox-active ketoamide moiety (*Ayikpoe and Latham, 2019*). In vivo, mycofactocins exist in oligoglycosylated forms (MFT-n), where n denotes the number of $\beta$–1,4-linked glucose moieties. The glycosylation requires the presence of the glycosyltransferase MftF (*Peña-Ortiz et al., 2020a*). In some mycobacteria, the glycosyl chain of MFT-n is further modified by methylation at the second sugar moiety (yielding MMFT-n) by the mycofactocin-associated methyltransferase MftM (*Ellerhorst et al., 2022*).

The reduction of M/MFT-n to M/MFT-nH$_2$ during growth on ethanol is likely to be catalyzed by the above-mentioned alcohol dehydrogenase Mno/Mdo (*Krishnamoorthy et al., 2019*; *Dubey and Jain, 2019a*). However, two important questions remained open concerning the mycofactocin system. First, it is unclear how MFT-nH$_2$ is re-oxidized to MFT-n after reduction by Mno/Mdo or other enzymes (*Figure 1B*). Such a step would be crucial to regenerate the cofactor and allow for further electron transfer to terminal electron acceptors such as oxygen. Second, the role of the *mftG* gene (*Figure 1A*), the terminal gene in the *mft* gene cluster of mycobacteria, encoding a flavoenzyme of the glucose-methanol-choline (GMC) oxidoreductase superfamily remained unknown. GMC oxidoreductases are present in bacteria, fungi, plants, and insects (*Sützl et al., 2019*; *Cavener, 1992*) and are known to catalyze redox reactions using a variety of different types of substrates and electron acceptors. Prominent examples are the fungal glucose oxidases, which are used in portable glucose

**Figure 1.** The mycofactocin redox system. (**A**) Schematic representation of the *mft* gene cluster of *M. smegmatis*. *mftA-F:* MFT biosynthetic genes. *mftR*: TetR-like regulator. *mftG*: GMC oxidoreductase (subject of this study). (**B**) Chemical structures of MMFT-n (oxidized methylmycofactocin) and MMFT-nH$_2$ (reduced form) and hypothetical scheme of MFT reduction by the ethanol dehydrogenase Mdo/Mno. The proposed mycofactocin dehydrogenase MftG is the subject of this study. X: Unknown electron acceptor.

biosensors for diabetes patients (*Ferri et al., 2011*). Other important representatives are the alcohol oxidases from the methylotrophic yeast *Pichia pastoris*, which uses oxygen as an electron acceptor (*Koch et al., 2016*), or the choline dehydrogenase of *E. coli*, which feeds electrons into the electron transport chain (*Landfald and Strøm, 1986*). We therefore hypothesized that MftG may be an oxidoreductase involved in mycofactocin-related metabolism (*Haft, 2011*). Here, we show that *mftG* (MSMEG_1428), encoding a flavoenzyme of the GMC family, is crucial for ethanol metabolism of the model organism *Mycolicibacterium smegmatis*. We further present evidence that MftG acts as a mycofactocin dehydrogenase and promotes MFT regeneration via electron transfer to the respiratory chain of mycobacteria.

## Results

### Phylogeny and co-occurrence of *mftG* and mycofactocin

Primary sequence alignments with known members of the GMC family and AlphaFold (*Varadi et al., 2022*; *Jumper et al., 2021*) prediction of the tertiary structure of MftG (WP_014877070.1) from *M. smegmatis* MC[2] 155 (*Figure 2A*) confirmed that MftG is a protein of the GMC superfamily with an intact FAD binding pocket and a conserved active site histidine (*Aleksenko et al., 2020*). While these properties are general characteristics of the GMC superfamily, a hallmark of the MftG subfamily of GMC oxidoreductases is their tight genetic linkage to the mycofactocin biosynthetic gene cluster *mftA-F* (*Haft, 2011*). These MFT-associated GMC homologs are defined by three protein families. The first, TIGR03970.1 (dehydrogenase, Rv0697 family) describes proteins from various actinobacteria and includes the *M. smegmatis* homolog. The other two families are TIGR04542 (GMC_mycofac_2) and NF038210.1 (GMC_mycofac_3). TIGR04542 is specific to the *Gordonia* genus, while NF038210.1 is exclusive to the *Dietzia* genus (*Haft, 2011*).

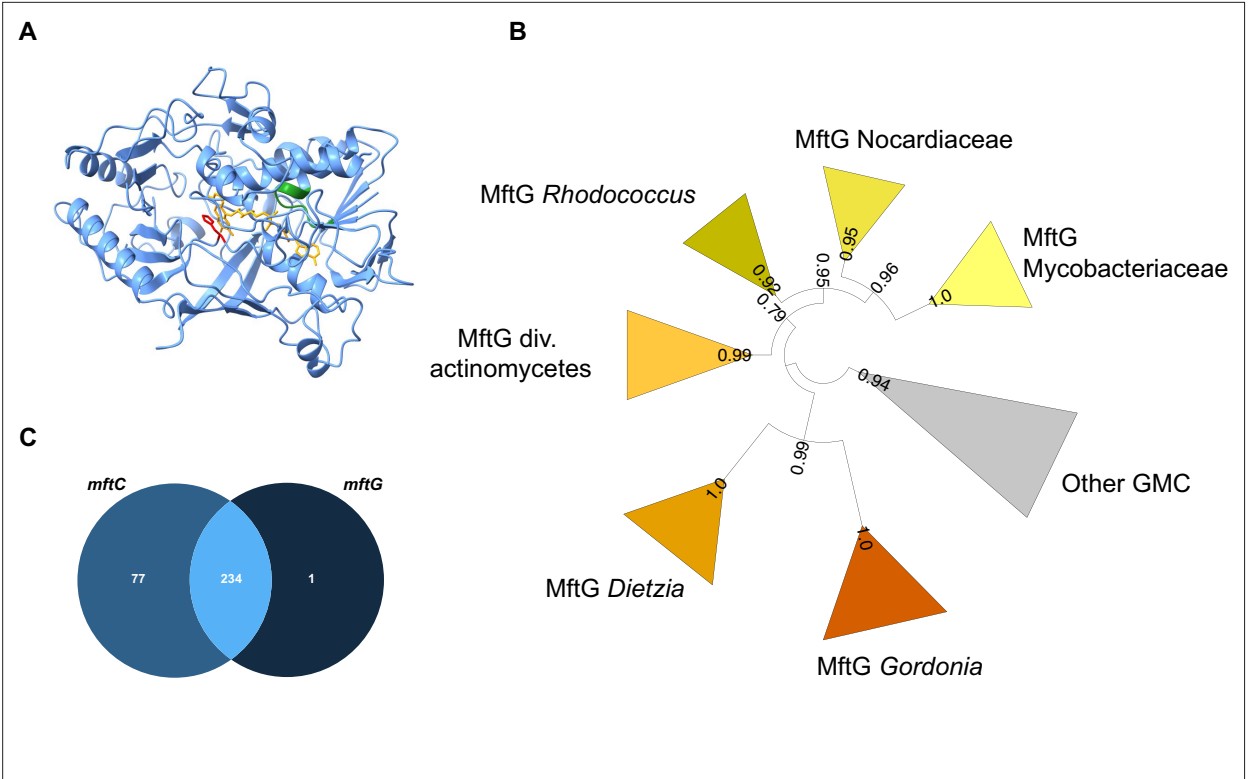

**Figure 2.** Bioinformatics analysis of MftG. (**A**) Structural model of MftG from *M. smegmatis* retrieved from the Alphafold database (**Gao et al., 2021**) with the FAD prosthetic group (yellow) modeled into the structure. Green: Rossman fold motif (GxGxxG), red: active site histidine (His411). (**B**) Collapsed phylogenetic tree (maximum likelihood) of GMC enzymes showing major MftG subfamilies. FastTree support values are shown on branches. The full tree is provided as *Figure 2—figure supplement 1* (**C**) Venn diagram representing the frequency of co-occurrence of *mftC* (left-medium blue) and *mftG* (right-dark blue) genes in 312 organisms that encode the MFT gene locus or MftG-like proteins.

The online version of this article includes the following figure supplement(s) for figure 2:

**Figure supplement 1.** Phylogenetic tree (maximum likelihood) of the MftG and GMC sequences described in *Supplementary file 2, table S1*.

To further investigate the co-occurrence of *mftG* with the *mft* gene cluster we retrieved a set of genomes encoding MftG or MftC homologs, the latter serving as a proxy for the *mft* biosynthetic gene cluster (*Supplementary file 2, table S1*). To refine the annotation of MftGs, we first performed a phylogenetic analysis. To the MftG candidates, we added further GMC-superfamily enzymes from the same genome set as well as sixteen experimentally characterized GMC enzymes used in a previous study (*Aleksenko et al., 2020*). Maximum Likelihood analysis (*Figure 2B*, *Figure 2—figure supplement 1*) clustered the MftG candidates described by TIGR03970 into a well-supported clade, which comprised MftG proteins from *Rhodococcus*, other *Nocardiaceae* and *Mycobacteriaceae*. The *Gordonia* and *Dietzia* MftG candidates were also phylogenetically related to the main MftG clade. We therefore defined all sequences belonging to this clade as MftG. Two sequences previously annotated as MftG, however, were placed outside of the proposed MftG clade. Their corresponding genomes (*Nocardia terpenica* and *Peterkaempfera bronchialis*) did not contain any MftC candidate and the MftG candidates were therefore treated as misannotations. We conclude that all MftG candidates are monophyletic. To further improve the annotations, the gene neighborhood of the *mftG* candidates was investigated, confirming that *mftG* candidates are frequently located within a 5 kb distance from *mft* genes (*Supplementary file 2, table S1*). After refinement of the MftG annotation, we proceeded with co-occurrence analysis of *mftC* and *mftG* in our microbial genome set. In a total of 312 genomes that contained either *mftC* or *mftG*, 311 harbored an *mftC* homolog, and 235 a putative *mftG* homolog. In 234 genomes the two genes co-occurred (*Figure 2C*). Only one genome (*Herbiconiux* sp. L3-i23) encoding a *bona fide mftG* did not harbor any *mftC* homolog. However, close inspection revealed the presence of *mftD*, *mftF*, and a potential *mftA* gene but a loss of *mftB, C and E* in this organism. This result reinforced the hypothesis that MftG enzymes strictly require mycofactocin. On the other

hand, about 25% of the genomes encoding MftC lack an MftG enzyme. It remains an open question for future investigations, whether other enzymes can complement the function of MftG or whether the function of MftG is dispensable in these organisms.

## The role of MftG in growth and metabolism of mycobacteria

To investigate the physiological role of MftG, a *mftG* deletion mutant (Δ*mftG*) was generated in *M. smegmatis* MC[2] 155. Since *mft* mutant strains typically display defects in ethanol utilization (**Krishnamoorthy et al., 2019**), we compared the growth of *M. smegmatis* MC[2] 155 Δ*mftG* with the WT (wild-type) strain on media containing 10 g L[−1] of ethanol as the sole carbon source (**Figure 3A**). Indeed, almost no growth of Δ*mftG* was detected on ethanol (**Figure 3A**), whereas the growth curve of the Δ*mftG* mutant on glucose-containing media was indistinguishable from the isogenic WT strain (**Figure 3B**).

We also investigated the growth of the WT and the Δ*mftG* strain on several related carbon sources and recorded growth curves (**Figure 3—figure supplement 1**). The growth of WT and mutant strains was not supported by 10 g L[−1] methanol, 5 g L[−1] hexanol, 0.01 g L[−1] acetaldehyde as sole carbon sources as previously reported (**Krishnamoorthy et al., 2019**). Notably, WT and Δ*mftG* cells displayed significant growth with 5 g L[−1] acetate or 10 g L[−1] glycerol. However, evaluation of the growth using other alcohols showed differential behavior. A prolonged lag phase of the Δ*mftG* strain was detected on 1-propanol (29 h), 1-butanol (18 h), and 1,3-propanediol (20 h) compared to the WT strain. Interestingly, a putative 1,3-propanediol dehydrogenase (MSMEG_6239) is present in the same operon as MSMEG_6242, indicating that 1,3-propanediol degradation might be also MFT-dependent (**Dubey and Jain, 2019b**). We further assessed the growth of Δ*mftG* using a panel of phenotype microarrays. While no further defect regarding carbon source utilization was detected, the Δ*mftG* mutant showed the ability to grow to a small extent on formic acid in contrast to the WT. Besides the different usage of carbon sources, only minor differences in bacterial growth were found in some of the sensitivity plates (**Figure 3—figure supplement 2**) demonstrating that the lack of the *mftG* gene did not induce relevant sensitivity towards antibiotics and stressors compared to WT when grown on glucose alone.

## The role of Δ*mftG* in ethanol metabolism of mycobacteria

Since the utilization of ethanol was the process that was most strikingly impacted by *mftG* inactivation, it was investigated further. First, the complementation of the Δ*mftG* deletion using an integrative vector carrying the *mftG* gene (Δ*mftG attB*::pMCpAINT-*mftG*, here named Δ*mftG-mftG*) was performed and resulted in the restoration of growth on 10 g L[−1] ethanol. Intriguingly, the growth curve of the complemented strain, which could be dysregulated in *mftG* expression, displayed a shorter lag phase, a faster second exponential growth phase, and a higher final biomass yield compared to the WT (**Figure 3A**). Altered growth kinetics of complement mutants on ethanol are a known phenomenon, albeit not always well understood (**Kpebe et al., 2023**). However, duplication of the *mftG* gene using the same vector in the WT strain (WT *attB*::pMCpAINT-*mftG*, here named WT-*mftG*) also showed enhanced growth. These results indicate that MftG might catalyze a rate-limiting step during ethanol utilization.

To further characterize the metabolic processes during ethanol consumption, the amount of ethanol and acetic acid present in the growth media with and without mycobacterial strains (WT, Δ*mftG*, Δ*mftG-mftG*, and WT-*mftG*) was measured (**Figure 3C and E**). As reported previously, ethanol consumption during exponential growth of WT mycobacteria was accompanied by acetic acid production (**Figure 3E**), a process known as overflow metabolism where the rate of ethanol oxidation to acetate exceeds the rate of carbon assimilation for biomass formation during fast growth (**Peña-Ortiz et al., 2020a**; **Peña-Ortiz et al., 2020b**). Although the Δ*mftG* strain could achieve residual growth after 45 h, the consumption of ethanol and the production of acetic acid remained at a very low level. (**Figure 3C and E**). While this finding is in line with the occurrence of a metabolic roadblock in the Δ*mftG* mutant, it indicates, however, that enzymatic activities supporting ethanol oxidation to acetate are not completely abolished in Δ*mftG* mutants. This interpretation is also supported by growth curves recorded on a combination of ethanol and glucose, where a simultaneous feeding enhanced the growth yield of the Δ*mftG* mutant (**Figure 3D**) suggesting at least partially intact ethanol utilization in Δ*mftG* cells.

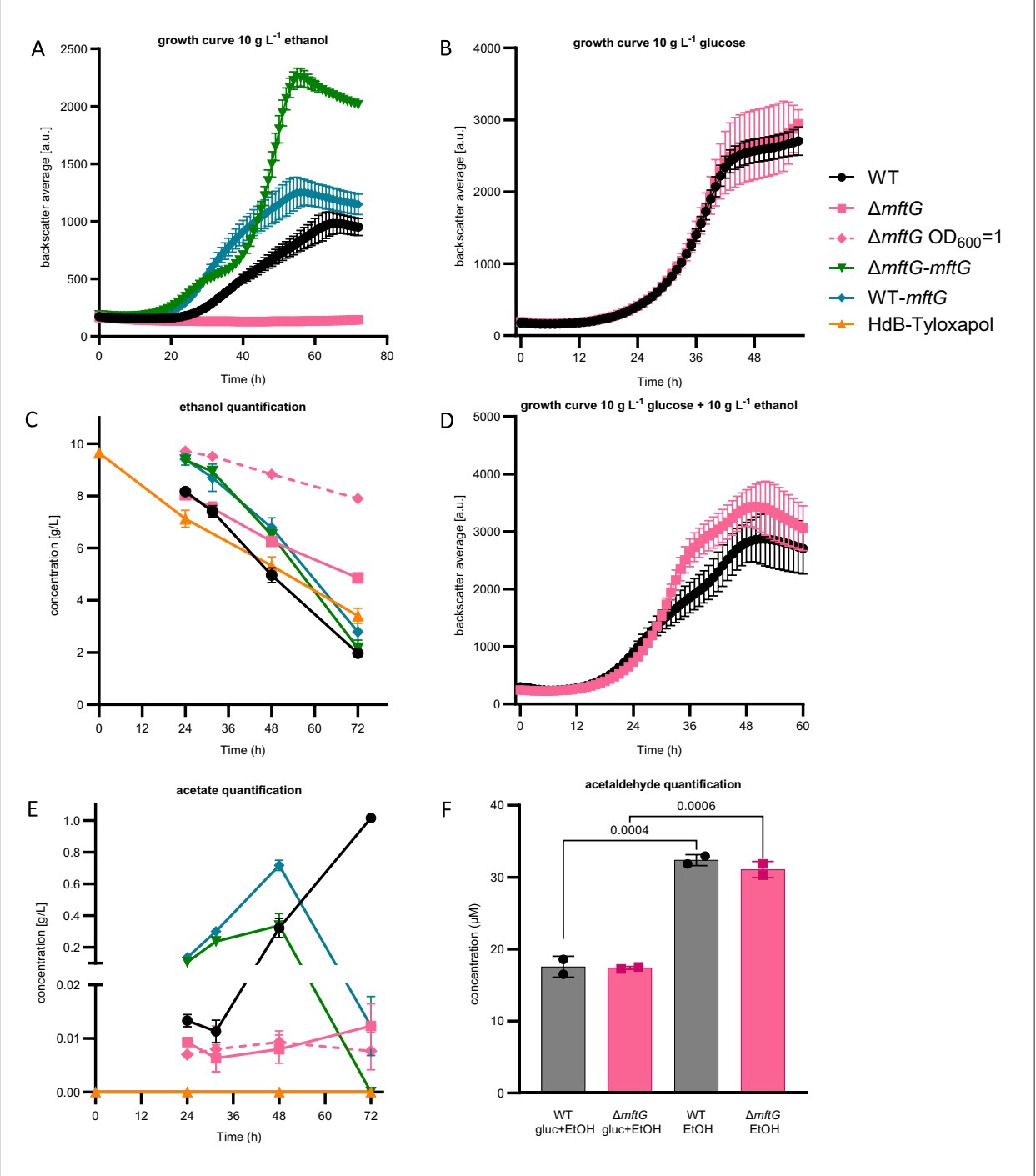

**Figure 3.** Effect of *mftG* gene deletion on mycobacterial ethanol metabolism. (**A**) Growth curve of *M. smegmatis* WT, Δ*mftG*, Δ*mftG-mftG*, and WT-*mftG* growing in HdB-Tyl with 10 g L$^{-1}$ ethanol as the sole carbon source. (**B**) Growth curve of WT and Δ*mftG* growing on 10 g L$^{-1}$ glucose (**C, E**). Ethanol and acetic acid quantification over time in *M. smegmatis* WT, Δ*mftG*, Δ*mftG-mftG*, and WT-*mftG* cultures in HdB-Tyl with 10 g L$^{-1}$ of ethanol and in uninoculated media as control. (**D**) Growth curve of WT and Δ*mftG* on 10 g L$^{-1}$ glucose and 10 g L$^{-1}$ ethanol combined. (**F**) Acetaldehyde quantification in culture supernatants of the WT and Δ*mftG* strains grown with 10 g L$^{-1}$ glucose and/or 10 g L$^{-1}$ ethanol. (●) *M. smegmatis* WT; (■) *M. smegmatis* Δ*mftG* mutant; (♦-dashed) *M. smegmatis* Δ*mftG* mutant grown with starting OD$_{600}$ 1; (▼) *M. smegmatis* Δ*mftG-mftG* complementation mutant; (♦) *M. smegmatis* double presence of the *mftG* gene; (▲) Medium with 10 g L$^{-1}$ of ethanol without bacterial inoculation. Measurements were performed in biological replicates, (growth curves: n≥3, ethanol and acetate quantification: n=3). Error bars represent standard deviations. Statistical analysis was performed with ordinary one-way ANOVA with Tukey's multiple comparison test, p-values depicted in the figure.

The online version of this article includes the following figure supplement(s) for figure 3:

*Figure 3 continued on next page*

*Figure 3 continued*

**Figure supplement 1.** Effect of *mftG* gene deletion on mycobacterial growth using different carbon sources.

**Figure supplement 2.** Differences between the relative growth of the *M. smegmatis* WT (black) and Δ*mftG* (pink) strains using the phenotypic arrays.

Typically, ethanol is oxidized to acetaldehyde first, which is further oxidized to acetic acid. To test whether the metabolic roadblock affects acetaldehyde formation or consumption, we also quantified acetaldehyde from supernatants of WT and Δ*mftG* cells grown on ethanol or ethanol combined with glucose (*Figure 3F*). Interestingly, when comparing WT and the Δ*mftG* mutant, we noticed that both strains produced a similar amount of acetaldehyde regardless of whether ethanol was used as the sole carbon source or supplemented (*Figure 3F*), indicating that acetaldehyde formation is not altered in the mutant strain. At this point, it may be helpful to revisit the fact that Δ*mftG* mutants grew well on acetate (*Figure 3—figure supplement 1*). This observation together with the detection of acetaldehyde and acetate in the supernatants of Δ*mftG* cultures excludes the hypothesis that *mftG* is required for acetaldehyde and acetate assimilation.

When investigating the carbon metabolism of the Δ*mftG-mftG* (complement) and WT-*mftG* (overexpression) strains, an inverted phenotype became visible. Parallel to the accelerated and enhanced growth described above (*Figure 3A*), the overexpression strains displayed higher rates of ethanol consumption as well as an earlier onset of acetate overflow metabolism and acetate consumption (*Figure 3E*). These results indicate that MftG activity is, at least indirectly, related to ethanol oxidation. Notably, the accelerated turnover of the volatile substrate ethanol, which is subject to substantial evaporation during the cultivation process, to acetate, which is less volatile, could explain the enhanced final growth yield of the complement and overexpression strains.

## The impact of ethanol on survival and cell division of Δ*mftG* mutants

To investigate whether the reduced growth of Δ*mftG* mutants on ethanol is due to limited carbon and energy supply or rather a consequence of insufficient ethanol detoxification, we cultivated bacteria on combinations of glucose and ethanol and recorded growth curves. Surprisingly, simultaneous feeding with 10 g L$^{-1}$ glucose and 10 g L$^{-1}$ ethanol even promoted the growth of the Δ*mftG* mutant. In addition, we determined the percentage of dead cells upon cultivation with different carbon sources using propidium iodide staining followed by flow cytometry quantification (*Figure 4A*). Regardless of the genotype, the bacterial cultures grown on glucose or ethanol as the sole carbon source and under starvation displayed a similar percentage of dead cells after a cultivation period of 72 h. We, therefore, conclude that the inability of Δ*mftG* cells to grow on ethanol alone is not due to ethanol toxicity. This conclusion was further supported by measurements of the transmembrane potential that did not reveal any disturbance of the proton motif force (PMF) in the mutant (*Figure 4B*). Previous studies of the transmembrane potential of *M. smegmatis* Δ*mftC* cells, another mutant unable to grow on ethanol as a carbon source, showed the same pattern with no differences compared to WT (**Krishnamoorthy et al., 2019**). However, unexpectedly, an elevated proportion of dead cells was detected when Δ*mftG* was cultivated on combined carbon sources, while the WT tolerated this condition well. This finding emphasizes the role of the mycofactocin system in enhancing the metabolic adaptability of mycobacterial cells under increasingly complex environmental conditions.

To reveal potential morphological changes induced by ethanol treatment, we further examined the Δ*mftG* mutant incubated with different carbon sources by super-resolution microscopy. The identification of elongation and replication sites was achieved by the sequential incubation of the cells with the fluorescent D-amino acid labeling probes NADA (green) and RADA (red; *Figure 4C–E*). These probes are incorporated via extracellular routes of mycobacterial peptidoglycan assembly, labeling both elongation poles as well as the sidewall (**García-Heredia et al., 2018**). The *M. smegmatis* WT and Δ*mftG-mftG* cultivated on glucose as the sole carbon source appear elongated with mainly polar growth (accumulation of RADA at the poles indicates the most recent active site of peptidoglycan incorporation). NADA occurs mainly at a non-polar location (previous active replication site). The Δ*mftG* mutant grown on glucose, however, showed few yellow cells (overlap of green and red) indicating growth arrest on a small part of the culture. Some cells are not actively synthesizing peptidoglycan and thus not growing, which leads to the overlap of both dyes (*Figure 4C*).

Mycobacteria grown on glucose and ethanol showed a regular dispersion profile of both dyes, with polar RADA, non-polar NADA, and occasional mid-cell septum formation with no obvious differences

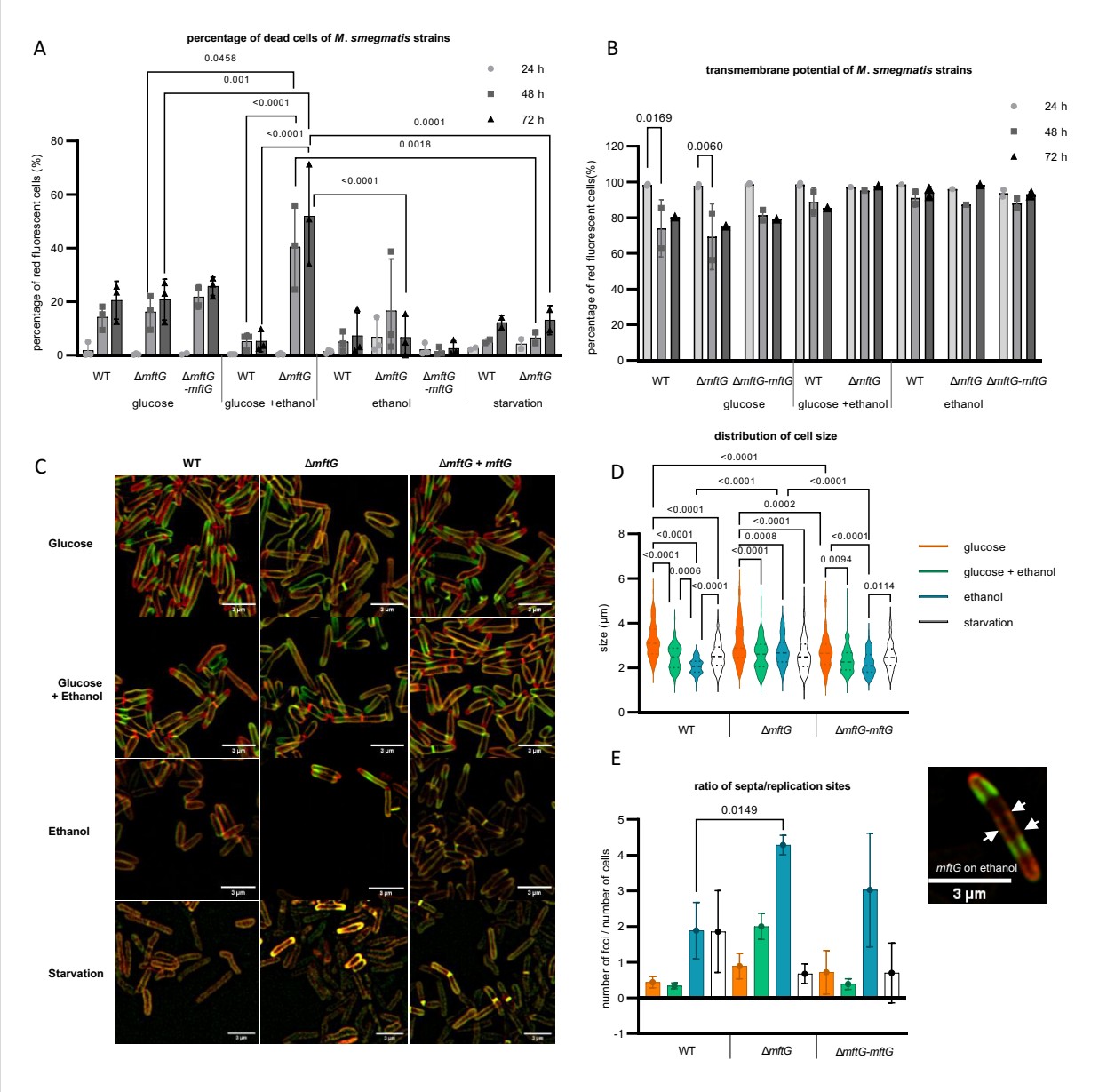

**Figure 4.** Phenotypic characterization of mycobacterial strains grown on HdB-Tyl with glucose and/or ethanol or starvation. (**A**) Quantification of dead cells by flow cytometry using propidium iodide of *M. smegmatis* strains grown throughout 72 hr. Biological replicates, starvation cells n=2, other conditions n=3. (**B**) Quantification of cells with normal transmembrane potential by flow cytometry of the *M. smegmatis* cultures throughout 72 h. Biological replicates n=3. (**C**) Super-resolution microscopy images of *M. smegmatis* strains at exponential phase or 48 h of starvation, labeled with NADA (green), RADA (red), superposition of NADA and RADA (yellow). Bar size: 3 μm. (**D**) Cell size distribution obtained from super-resolution microscopy of the *M. smegmatis* strains at exponential phase or 48 h of starvation. (**E**) Ratio of the number of replication sites to the number of cells of the *M. smegmatis* strains cultures at exponential phase or 48 h of starvation, together with a microscopy image of a single Δ*mftG* cell at 48 h grown on ethanol, with arrows pointing to the several septa stained with NADA (green) and RADA (red). Bar size: 3 μm. Color legend: (**A,B**): ● – sample at 24 h; ■– sample at 48 h; ▲– 72 h. (**C**) (**D,E**): orange – 10 g L⁻¹ glucose; green – 10 g L⁻¹ glucose and 10 g L⁻¹ ethanol; blue – 10 g L⁻¹ ethanol; white – starvation for 48 h. Statistical analysis was performed for PI, cell size and ratio of replication sites per cell with ordinary one-way ANOVA, for transmembrane potential with ordinary two-way ANOVA, all using Tukey's multiple comparison test. The p-values are depicted on the figure, microscopy-based analysis performed with technical replicates (n=3). Error bars represent standard deviations.

between the three genotypes. In contrast, cultivation on 10 g L⁻¹ ethanol alone as well as under starvation condition (no carbon source) had a strong influence on peptidoglycan synthesis as the dyes show decreased incorporation into the cell wall. The simultaneous incorporation of both dyes in WT and Δ*mftG-mftG* cells grown on ethanol as the sole carbon source reflected the reduced growth rate,

as shown by the growth curves (*Figure 3A and B*) compared to the glucose condition. Interestingly, Δ*mftG* mutant cells struggled to divide on ethanol displaying multiple septa and extreme elongation compared to WT on ethanol, leading to a higher ratio of foci per cell compared to WT grown on the same condition (*Figure 4D and E*). Independent of the genotype, the cultivation in the presence of ethanol significantly decreased cell size. A similar phenotype was previously observed in *Mycobacterium vaccae* cells exposed to ethanol (*Pacífico et al., 2018*). Starving cells were also significantly reduced in size in WT and Δ*mftG* strains compared to growth on glucose alone (*Figure 4C and E*), reflecting the limitation of resources available for propagation and elongation. Especially under starvation conditions, the cells appeared predominantly yellow, suggestive of growth arrest. A previous study on the effect of starvation on mycobacteria also showed the formation of large cells with multiple septa as a hallmark of starvation, from which after 14 days, mildly starved cells (with traces of carbon source available) remodeled into small resting cells (*Wu et al., 2016*). Our results indicated that the major phenotype of Δ*mftG* on ethanol as a sole source of carbon is a growth defect comparable to a starvation effect. When incubated solely with this carbon source, Δ*mftG* cells are almost unable to divide, but remain alive and in a metabolically active state, most likely consuming ethanol in trace amounts.

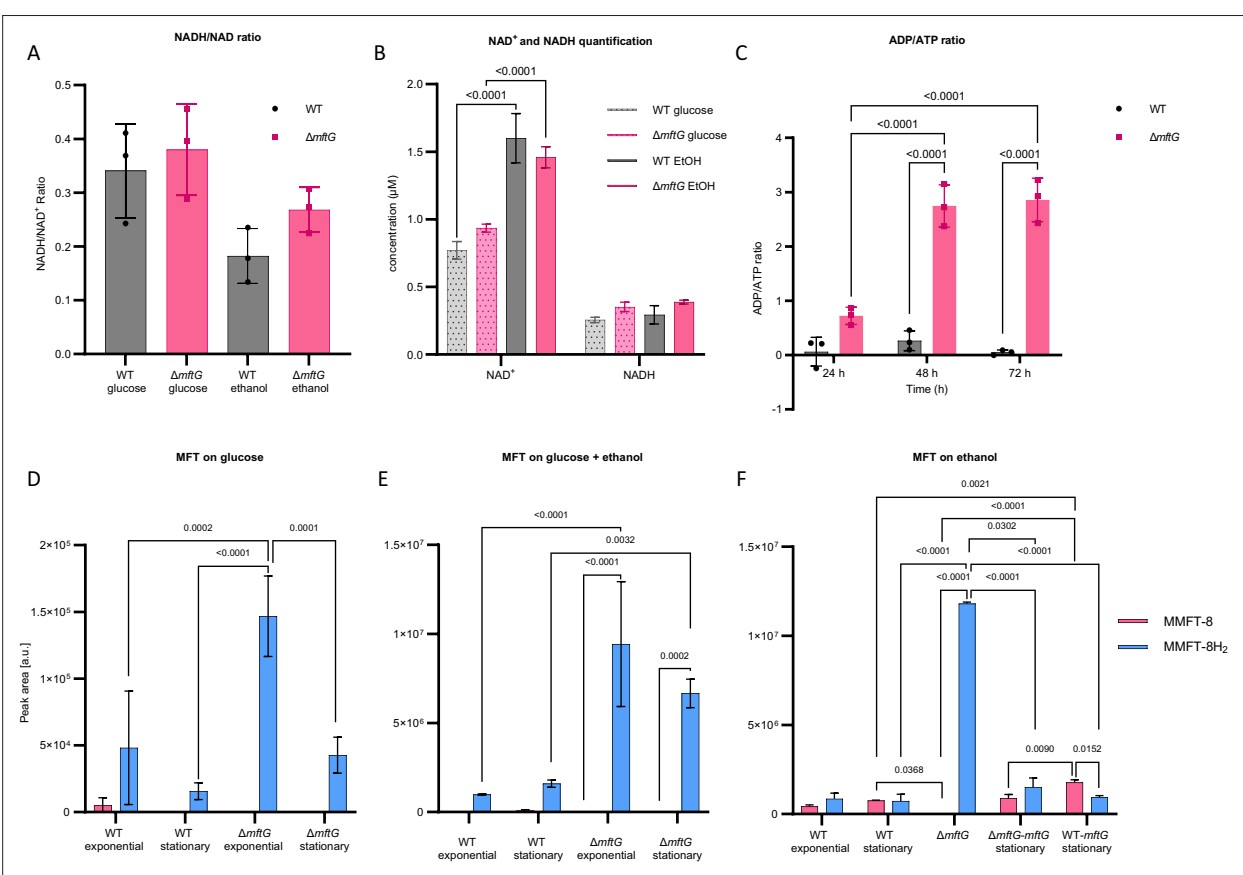

**Figure 5.** Cofactor metabolism of *M. smegmatis* strains. (**A**) NADH/NAD⁺ ratio of *M. smegmatis* WT and Δ*mftG* grown on HdB-Tyl with either 10 g L⁻¹ glucose or 10 g L⁻¹ ethanol at exponential phase. (**B**) NADH and NAD⁺ quantification of *M. smegmatis* WT and Δ*mftG* grown on HdB-Tyl with either 10 g L⁻¹ glucose or 10 g L⁻¹ ethanol at exponential phase. (**C**) ADP/ATP ratio of *M. smegmatis* WT and Δ*mftG* grown on HdB-Tyl with 10 g L⁻¹ ethanol at 24 hr, 48 h and 72 h. (**D, E, F**) Targeted comparative metabolomics of *M. smegmatis* WT, Δ*mftG*, Δ*mftG-mftG,* and WT-*mftG* strains. The most representative MFT species, methylmycofactocinone with 8 glucose moieties (MMFT-8H₂, sum formula: C₆₂H₉₉NO₄₃, RT: 6.82 min, *m/z* 1546.5665 [M+H]⁺) and methylmycofactocinol with 8 glucose moieties (MMFT-8, sum formula: C₆₂H₉₇NO₄₃, RT: 7.18 min, *m/z* 1544.5507 [M+H]⁺), was used to reflect MFT obtained from *M. smegmatis* strains. The bacteria were grown in HdB-Tyl with either (**D**) 10 g L⁻¹ glucose, (**E**) 10 g L⁻¹ ethanol, or (**F**) 10 g L⁻¹ glucose combined with 20 g L⁻¹ ethanol. Samples of the different growth phases are represented in the chart. A sampling at 60 h of Δ*mftG* was chosen to sample the residual growth of the strain on ethanol as the sole carbon source. Statistical analysis was performed with one- or two-way ANOVA with Dunnett's multiple comparison test for NADH/NAD⁺ ratio and, Tukey's test for the rest, with most relevant p-values depicted on the figure. Measurements were performed in biological replicates (n=3), error bars represent standard deviations.

## Impact of *mftG* on mycobacterial cofactor metabolism

Based on the results above, we concluded that MftG is crucial for proper ethanol metabolism in mycobacteria. However, deletion of its corresponding gene still allowed for basal metabolic activity and ethanol oxidation. One way to explain these findings is that MftG is involved in cofactor regeneration during growth on ethanol as a carbon source. The genetic linkage of *mftG* with mycofactocin biosynthesis supported the hypothesis that MftG might be responsible for mycofactocin regeneration. However, before directly addressing mycofactocin metabolism, we decided to monitor the central respiratory redox cofactor NAD. We therefore determined the NADH/NAD$^+$ ratios of WT and knockout during the exponential phase when bacteria were grown on glucose or ethanol as the sole carbon sources. However, the presence or absence of *mftG* did not significantly influence the NADH/NAD$^+$ ratio of bacteria grown on either carbon source (*Figure 5A*). This finding indicated that NAD homeostasis remained intact in the Δ*mftG* mutant. Notably, when looking at absolute NAD$^+$ levels, (*Figure 5B*) an increase of the NAD pool in cells growing on ethanol was detected, however, independent of the genotype. Along with central redox (NAD$^+$) metabolism, we also monitored the central energy metabolism of the cells by determining ADP/ATP ratios. These clearly supported the hypothesis that Δ*mftG* cells suffered from starvation. The mutant strain exhibited a significant energy deficit at all three sampled time points, with the energy depletion progressively worsening between 24 and 48 h of incubation (*Figure 5C*).

Previous studies of *M. smegmatis* under starvation also showed a reduction of ATP levels as a result of hampered energy metabolism (*Wu et al., 2016*). It should be mentioned that ATP synthesis strictly depends on respiration in *M. smegmatis*. Mycobacteria typically cannot sustain growth, even on fermentable substrates, via substrate-level phosphorylation alone (*Tran and Cook, 2005*).

After confirming energy shortage but intact NAD homeostasis in Δ*mftG* cells, we decided to test whether MftG is involved in mycofactocin regeneration. To this end, we directly analyzed the MFT pool using targeted liquid chromatography-high resolution mass spectrometry (LC-MS). We have previously shown that the total pool size of MFT species is highly expanded when *M. smegmatis* uses ethanol as a sole carbon source and that both reduced (M/MFT-nH$_2$) as well as oxidized species (M/MFT-n) are stable when extracted and can thus be detected by LC-MS (*Peña-Ortiz et al., 2020a*). Here, we performed a comparative analysis of the metabolome of *M. smegmatis* WT and Δ*mftG* grown in HdB-Tyl supplemented with either 10 g L$^{-1}$ glucose, 10 g L$^{-1}$ ethanol, or both carbon sources combined (10 g L$^{-1}$ glucose and 20 g L$^{-1}$ ethanol). Metabolites were extracted during the exponential or stationary growth phase. Additionally, metabolome extracts from complemented Δ*mftG-mftG* and WT-*mftG* strains grown on HdB-Tyl with 10 g L$^{-1}$ ethanol sampled only at the stationary phase were analyzed. Strikingly, this analysis revealed significantly elevated levels of MMFT-8H$_2$ in Δ*mftG* compared to the other strains tested (*Figure 5D–F*). MMFT-8H$_2$ is the major representative of reduced mycofactocins (mycofactocinols) in *M. smegmatis*. Interestingly, Δ*mftG* strains contained almost none of the corresponding oxidized mycofactocinone (MMFT-8) in any of the conditions tested. These results strongly supported the hypothesis that MFT regeneration was hampered in the Δ*mftG* mutant while MFT reduction was still taking place, thus resulting in a near-total conversion of mycofactocinones to mycofactocinoles. This phenotype was also observed when Δ*mftG* cells were cultivated on glucose and ethanol combined (*Figure 5E*) and even on glucose alone (*Figure 5D*). Complementation of the Δ*mftG* mutant (Δ*mftG-mftG*) reverted the MMFTH$_2$ to MMFT ratio back to WT levels. In contrast, the overexpression strain WT-*mftG* grown in ethanol contained significantly more MMFT-8 (oxidized) compared to other strains in the same condition (*Figure 5F*), thus providing further evidence that MftG is involved in reoxidation of mycofactocin and re-enforcing that this step might even represent a rate-limiting step during ethanol utilization.

## MFT dehydrogenase assay with recombinant MftG

To further support the hypothesis that MftG is responsible for MFT reoxidation, purification of MftG as a recombinant fusion protein with hexahistidine tag followed by in-vitro enzyme assays was attempted. For homologous production of recombinant, hexahistidine-tagged MftG, *M. smegmatis* Δ*mftG* was chosen as an expression host (resulting genotype: Δ*mftG-mftG*His$_6$). The restoration of the growth on ethanol of the Δ*mftG-mftG*His$_6$ expression strain indicated that the recombinant protein was produced in an active form. Semi-purified cell-free extracts of Δ*mftG-mftG*His$_6$ incubated with metabolome extract from *M. smegmatis* Δ*mftG* as substrate showed that the *mftG-mftG*His$_6$ lysate

significantly increased the amount of MMFT-8 and decreased the amount of the MMFT-8H$_2$ while the Δ*mftG* lysate did not affect the ratio of the two compounds (*Figure 6A*). Inspired by these results we conducted assays with semi-purified cell-free extracts from *M. smegmatis* Δ*mftG-mftG*His$_6$ with higher amounts of purified cofactors as substrates. We added synthetic PMFTH$_2$ and purified MMFT-2H$_2$ as substrates as well as synthetic PMFT as a negative control. The negative control using PMFT contained trace amounts of PMFTH$_2$ from the start and no clear trend was observed over the time course of the reaction (*Figure 6B*). In contrast to this, we observed conversion of both reduced substrates PMFTH$_2$ (*Figure 6C*) and MMFTH-2H$_2$ (*Figure 6D*) to the corresponding oxidized species PMFT and MMFT-2, respectively. However, the reaction proceeded only with partial conversion of the substrate. Additionally, since some enzymes of the GMC family can utilize oxygen as electron acceptors, the role of oxygen in MftG activity was accessed. To this end, the assays were performed under a nitrogen atmosphere at <0.1% oxygen. Under these conditions, the MftG activity remained unchanged (*Figure 6C and D*), thus ruling out that MftG acts as a mycofactocin oxidase using O$_2$ as an electron acceptor.

In order to confirm that the activity detected in cell-free lysates was not due to background effects, the heterologous production of MftG in *Escherichia coli* followed by activity assays was attempted. Despite poor production and solubility of MftG in *E. coli*, we detected MftG activity in semi-purified fractions of *E. coli* producing MftG tagged with maltose-binding-protein (pPG36). While mycofactocin dehydrogenase activity was clearly linked to the overexpression of MftG (*Figure 6—figure supplement 1*), only partial turnover of substrate was still observed suggesting the absence of an appropriate electron acceptor. A screening for several potential electron acceptors was performed using MMFT-2H$_2$ as substrate. However, none of the potential acceptors tested, DCPIP, NAD$^+$, NDMA, and PMS, were able to increase substrate turnover (*Figure 6E*). Assays with increasing concentrations of protein suggested that the observed activity was possibly due to single-turnover reactions, most likely using the FAD cofactor in the active site as an electron acceptor (*Figure 6F*).

We, therefore, concluded that MftG can indeed interact with mycofactocins as electron donors but might require complex electron acceptors, for instance, proteins present in the respiratory chain.

## Influence of MftG and MFT in mycobacterial respiration

The phenotypes observed for cell growth and division together with an increased ratio of ADP/ATP in the Δ*mftG* strain suggested a potential involvement of MftG in mycobacterial respiration. To further test this hypothesis, the consumption of O$_2$ from whole cells of *M. smegmatis* WT and Δ*mftG* grown on HdB-Tyl with 10 g L$^{-1}$ of ethanol was measured in a respirometer equipped with a Clark-type electrode. This experiment revealed that the respiration rate of Δ*mftG* reached only about 45% of the WT level (*Figure 7*). Notably, the *M. smegmatis* carrying a duplicated *mftG* gene, WT-*mftG*, showed accelerated oxygen consumption, supporting the idea that MftG is a limiting factor in the respiration of mycobacteria grown on ethanol.

To test whether MftG might interact with components of the respiratory chain, we investigated mycobacterial membrane preparations for respiratory activities. The influence of mycofactocin on mycobacterial respiration was tested using isolated membranes from *M. smegmatis* WT supplemented with either NADH, succinate, synthetic PMFTH$_2$, or purified MMFT-2H$_2$. In these respiration assays, the consumption of oxygen after supplementation with NADH or succinate was highly increased showing the expected respiratory activity of the isolated membranes via NADH or succinate dehydrogenases (*Figure 7*). The addition of PMFTH$_2$ or MMFT-2H$_2$ stimulated oxygen consumption to a rate that was comparable to succinate-induced respiration of the WT strain. In *M. smegmatis*, cyanide is a known inhibitor of the cytochrome *bc1-aa3* but not of cytochrome *bd* (*Kana et al., 2001*). Therefore, the decrease of oxygen consumption in the presence of KCN (*Figure 7*) revealed that MFT-induced oxygen consumption is indeed linked to mycobacterial respiration.

The membrane fraction of the Δ*mftG* strain showed a decreased consumption of oxygen compared to WT. Although the auto-respiration, i.e., the consumption of oxygen in the absence of a stimulant, was lower in the Δ*mftG* strain, the supplementation of NADH and succinate stimulated oxygen consumption demonstrating that the membranes contained a functional electron transport chain. The addition of reduced mycofactocins to the reaction revealed strongly reduced respiration when compared to the WT membrane. To test whether increased O$_2$ consumption was linked to the oxidation of mycofactocinols, we also carried out LC-MS analysis of the samples used in the respiration assays. Indeed, mycofactocinols were oxidized to mycofactocinones (e.g. PMFTH$_2$ to PMFT) under

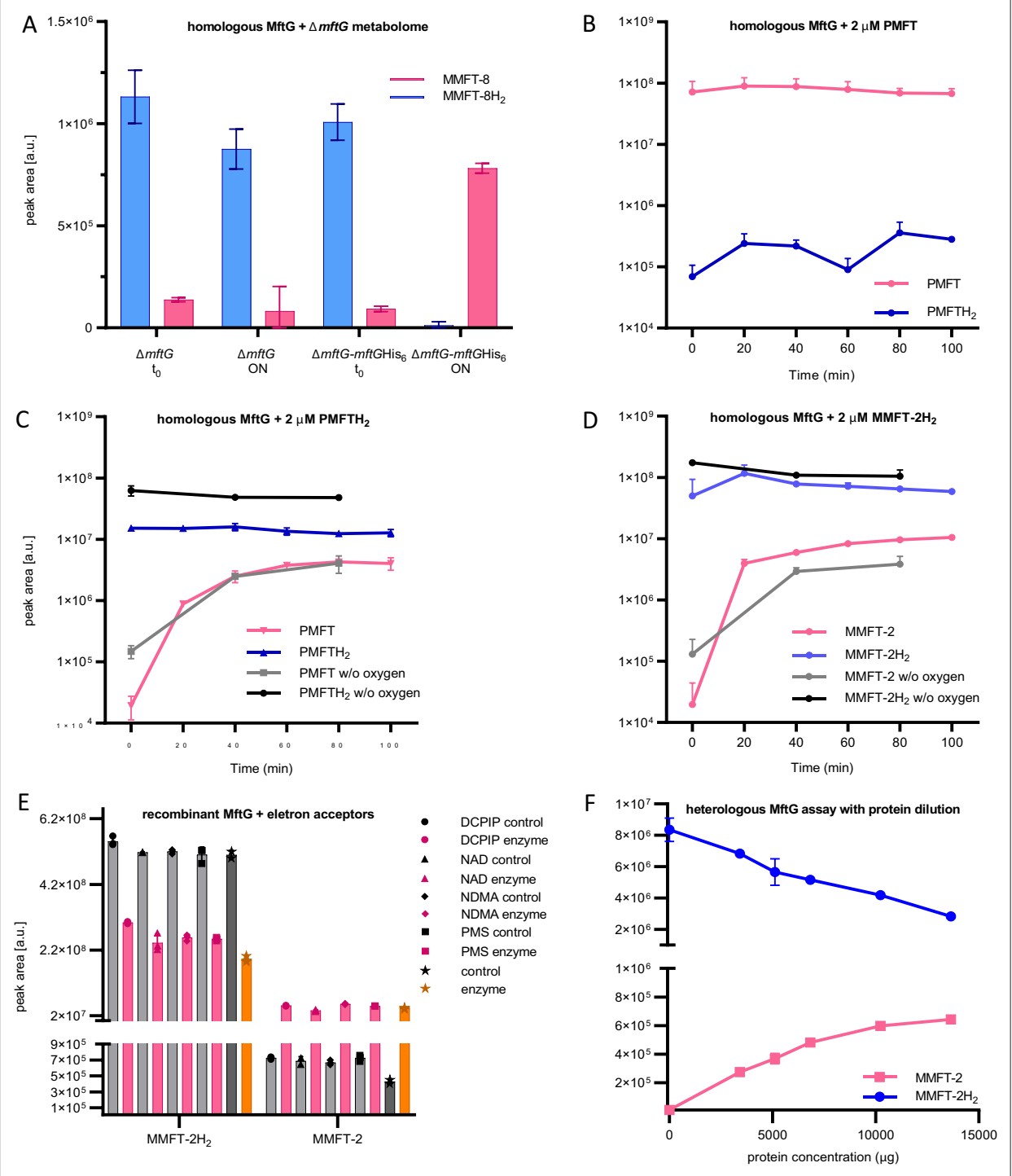

**Figure 6.** MftG assays with recombinant enzymes and MFTs as substrates. (**A**) Mycofactocinol oxidation assay with semi-purified cell-free extract of *M. smegmatis* Δ*mftG* or Δ*mftG-mftG*His$_6$ using Δ*mftG* metabolome extract as substrate (naturally enriched in reduced MFT as described in **Figure 5F**). Result showing the oxidation of MMFT-8H$_2$ to MMFT-8 after overnight incubation (ON) when semi-purified cell-free extract from Δ*mftG-mftG*His$_6$ is used. t$_0$ – start of the assay. (**B**) Assay with semi-purified cell-free extract of Δ*mftG-mftG*His$_6$ using synthetic PMFT as a control substrate showed no relevant reaction. (**C**) Successful oxidation of PMFTH$_2$ when synthetic PMFTH$_2$ was used as substrate (C$_{13}$H$_{17}$NO$_3$, RT: 7.84 min, *m/z* 236.1281 [M+H]$^+$) to PMFT (C$_{13}$H$_{15}$NO$_3$, RT: 8.40 min, *m/z* 234.1125 [M+H]$^+$). Black and grey lines depict reactions performed in an anaerobic chamber. (**D**) Successful oxidation of MMFT-2H$_2$ (C$_{26}$H$_{39}$NO$_{13}$, RT: 7.10 min, *m/z*: 574.2494 [M+H]$^+$) to MMFT-2 (C$_{26}$H$_{37}$NO$_{13}$, RT: 7.47 min, *m/z* 572.2338 [M+H]$^+$). Black and grey lines depict assays performed in the anaerobic chamber (**E**) Mycofactocinol (MMFT-2H$_2$) oxidation using MftG heterologously produced in *E. coli* and DCPIP, NAD$^+$, NDMA, and PMS as potential electron acceptors. Control – no MftG added; Enzyme – MftG added. (**F**) Dose-dependent effect of heterologously

*Figure 6 continued on next page*

*Figure 6 continued*

expressed MftG on the oxidation of MMFT-2H$_2$ to MMFT-2 was observed after a 24 h incubation period. Sample size of all experiments n=3. Error bars represent standard deviations.

The online version of this article includes the following figure supplement(s) for figure 6:

**Figure supplement 1.** Mycofactocin dehydrogenase activity of MftG and enzyme preparations obtained from *E. coli* BL21 (DE3).

active assay conditions, while only low amounts of PMFTH$_2$ were oxidized in membranes of the Δ*mftG* strain or by membranes treated with KCN. Two observations suggest a low level of MftG-independent mycofactocin oxidation: (I) The residual induction of oxygen consumption in the Δ*mftG* strain by mycofactocinols, (II) the formation of low amounts of oxidized mycofactocins in the Δ*mftG* strain (at a similar level to WT inhibited by KCN).

From the respirometry experiments, we conclude that MftG catalyzes the electron transfer from mycofactocinol to membrane-bound components of the respiratory chain. These could be the quinone pool as known from succinate dehydrogenase complex, or cytochromes. For example, MftG in mycobacteria might play a similar role as the subunit II of the pyrroquinoline quinone-dependent alcohol dehydrogenase (PQQ-ADH) complex from acetic acid bacteria. The subunit II transfers electrons from the alcohol dehydrogenase (subunit I/III) to the ubiquinone pool, from which further electron transfer to oxygen occurs (*Yakushi and Matsushita, 2010*).

## Transcriptomic analysis of *M. smegmatis* Δ*mftG* cells

To monitor genome-wide regulatory adaptation caused by *mftG* deletion, we employed high-throughput sequencing to record transcriptomic changes in WT and Δ*mftG* cells when grown to exponential phase (60 h of incubation for Δ*mftG* on ethanol) on glucose or ethanol. In the glucose condition, only nine out of 6506 annotated genes were classified as significantly upregulated in Δ*mftG* when compared to the WT (adjusted p-value <0.05 and log2FoldChange >2). Since these genes showed neither functional nor spatial clustering, we conclude that no meaningful differences in the mycobacterial transcriptome between WT and mutant grown on glucose were observed (*Supplementary file 3*, Table S2).

On the contrary, transcriptomics analysis of Δ*mftG* and WT grown on ethanol detected 578 genes as significantly up-regulated and 697 as down-regulated (adjusted p-value <0.05 and log2FoldChange >2 or <-2) in the mutant when compared to the WT (*Supplementary file 3* Table S2). The deletion of the *mftG* gene induced mainly processes related to transmembrane transporter activity, iron binding, monooxygenase activity, quinone activity and NADH dehydrogenase (*Figure 8A*). Downregulated processes were related to ATP binding and ATPase transmembrane movement, respiration, and DNA replication (*Figure 8B*). Significant changes, presumably due to energy limitation, took place in the Δ*mftG* mutant compared to the WT. For instance, the *nuo* genes (MSMEG_2050–2063) encoding NADH dehydrogenase I (NADH-I), were amongst the most highly expressed genes (*Figure 8*, *Supplementary file 3* Table S2). In mycobacteria, two NADH dehydrogenases (NDH-I and II) are active, while in *M. smegmatis* the NADH-I dehydrogenase typically contributes only a low proportion of total NADH activity (*Cook et al., 2014*). Notably, enhanced expression of the high-efficiency, proton-pumping NADH-I (*nou*) system was previously observed under energy-limited starvation conditions (*Berney and Cook, 2010*). In addition, alternative dehydrogenases fostering electron transfer from other electron donors to the respiratory chain were strongly upregulated (*Figure 8C*). These comprise enzymes like proline dehydrogenase (MSMEG_5117), pyruvate dehydrogenase (MSMEG_4710–4712), carbon monoxide oxidase (MSMEG_0744–0749), the hydrogenases 1 (*huc* operon, MSMEG_2261–2276) and 2 (MSEMG_2720–2719). Furthermore, quinone-dependent succinate dehydrogenase SDH1 (MSMEG_0416–0420) was highly upregulated when compared to the WT. Again, similar patterns were previously described in cultures growing under carbon limitation and can be seen as an attempt of the cell to compensate for poor loading of the electron transport chain (*Berney and Cook, 2010*). This finding is in line with our hypothesis that electron transfer to the respiratory chain is hampered and mutant cells suffer from mild starvation.

Reflecting the division arrest phenotype observed by super-resolution microscopy, peptidoglycan biosynthesis, represented by the genes *murABCDEFG* (e.g. MSMEG_1661, MSMEG_6276, MSMEG_2396, MSMEG_4194) was found to be downregulated. DNA replication was also affected

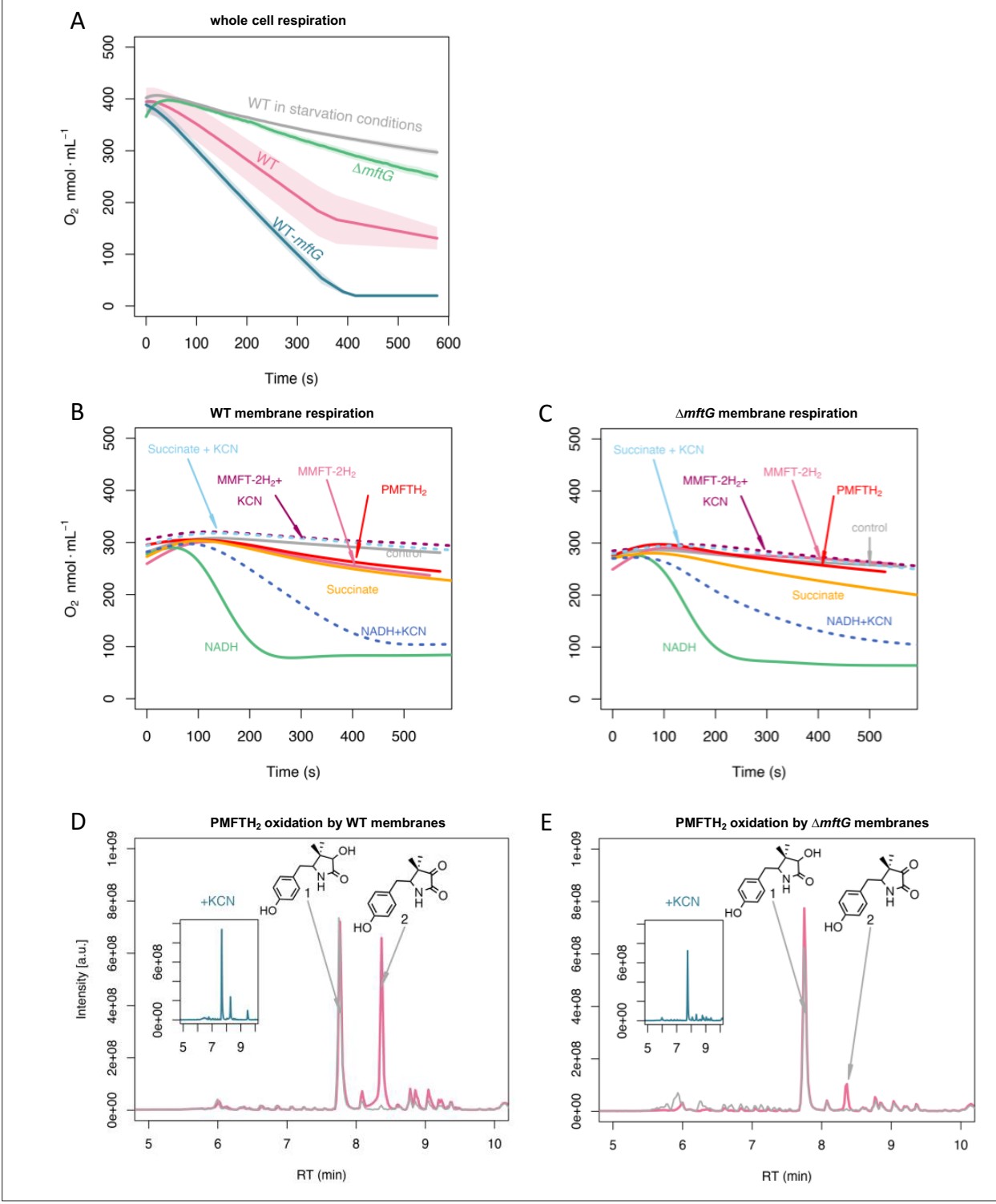

**Figure 7.** Respiratory activity (oxygen consumption) of *M. smegmatis* WT and Δ*mftG* mutants. (**A**) Respiration of intact cells. Average of n=3 (**B**) Respiration of WT isolated cell membranes and addition of electron donors as indicated in the figure. (**C**) Respiration of Δ*mftG* isolated cell membranes and addition of electron donors as indicated in the figure. NADH and succinate served as positive controls, and water as a negative control. KCN treatment served as inhibitor control. MMFT-2H$_2$ and PMFTH$_2$ were added to confirm MFT's role as an electron donor. (**D**) Oxidation of PMFTH$_2$ to PMFT in WT isolated membranes (combined LC-MS profiles). (**E**) Oxidation of PMFTH$_2$ to PMFT of the Δ*mftG* isolated membranes (combined LC-MS profiles). Each inset depicts the profile after KCN treatment. Representative data was selected from independent experiments n≥3.

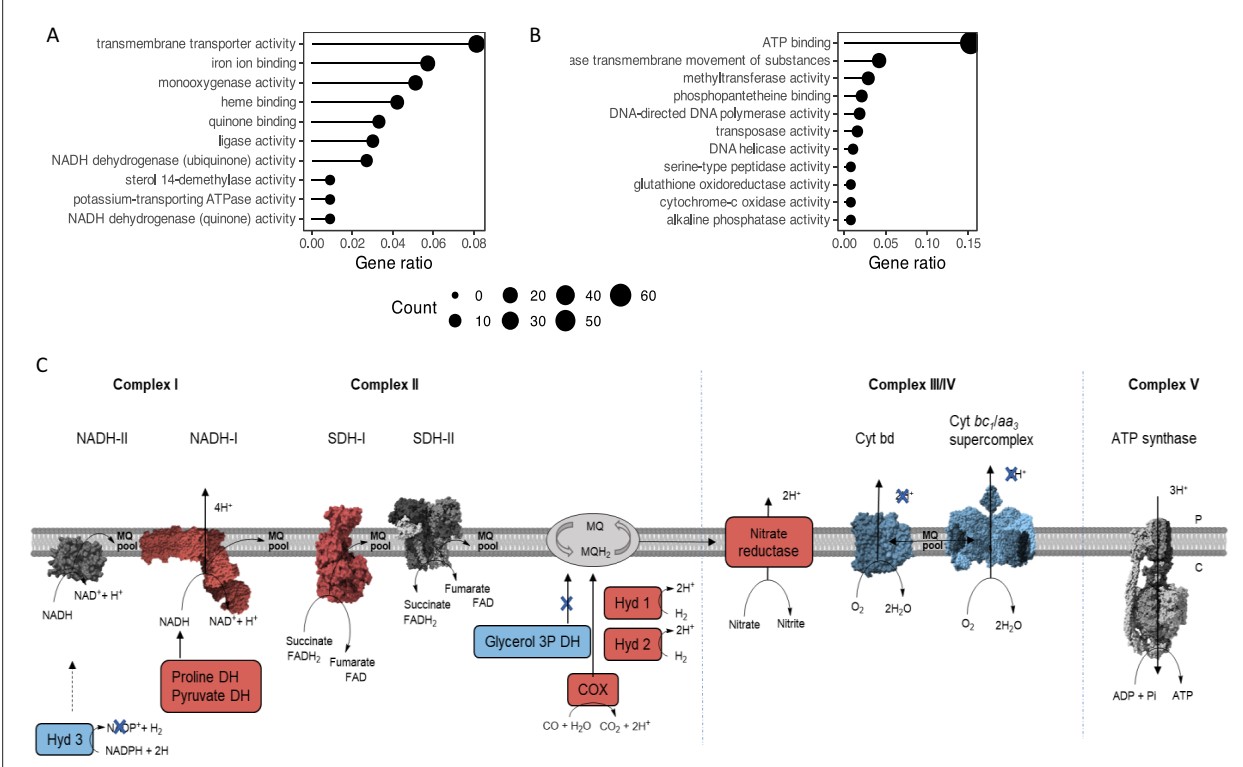

**Figure 8.** Representation of the main metabolic activities affected by *mftG* deletion in *M. smegmatis* grown on 10 g L$^{-1}$ ethanol as the sole carbon source compared to WT. (**A,B**) Functional annotation chart (Gene ontology enrichment analysis) of the (**A**) up- and (**B**) down-regulated processes. Gene ratio denotes the ratio of the involved genes (count) to the quantity of the genes making up the enriched terms. (**C**) Impact on the respiration of the mutant strain ΔmftG grown on 10 g L$^{-1}$ ethanol compared to WT strain. Blue represents genes downregulated p<0.05 and log2FC< –2. Red represents genes upregulated p<0.05 and log2FC >2. Protein figures retrieved from public databases NADH-II: A0QYD6, NADH-I: 8E9G, Cyt bd: 7D5I, Cyt bc1-aa3: 7RH5, SDH-II: 6LUM, Sdh-I: 7D6X, ATP synthase: 7NJK.

in ΔmftG grown on ethanol by the downregulation of RNaseH (MSMEG_4306), DNA polymerase (MSMEG_3839), helicase (MSMEG_6892), and all the modules of the DNA polymerase III holoenzyme (MSMEG_4259, MSMEG_3178, MSMEG_6285, MSMEG_6153, MSMEG_4572 and *dnaN*).

In general, the transcriptomic analysis of the ΔmftG strain on ethanol compared to WT in the same condition showed that the observed phenotype resembled energy-limiting conditions, accompanied by remodeling of the respiration chain to compensate for a lack of redox equivalents as well as inhibition of peptidoglycan synthesis and DNA replication.

## Discussion

Alcohols are widespread ingredients of various foods and beverages and are used as active principles of many disinfectants. Therefore, the impact of alcohols on commensal and clinically important bacterial strains has high relevance. However, ethanol metabolism in mycobacteria is still poorly characterized. While it is established that the cofactor mycofactocin plays a major role in this process, the fate of electrons during alcohol oxidation remained unknown (*Krishnamoorthy et al., 2019*; *Krishnamoorthy et al., 2021*).

Based on bioinformatics analysis, we hypothesized that MftG homologs, flavoenzymes of the GMC superfamily, might be involved in redox metabolism connected to MFT-dependent alcohol metabolism. Using *M. smegmatis* as a model system, we showed that MftG is indeed essential for ethanol metabolism, a process known to depend on both an intact mycofactocin biosynthetic pathway as well as the putative MFT-dependent alcohol-dehydrogenase Mdo/Mno. Mutants lacking the *mftG* gene incubated on ethanol as the sole carbon source displayed strongly impaired growth. Super-resolution microscopy combined with peptidoglycan labeling, transcriptomics as well as direct measurements

of ADP/ATP ratios demonstrated that mutants grown on ethanol suffer from energy limitation and growth arrest similar to a mild starvation condition. However, the mutant maintained a level of functional metabolism, conserving its transmembrane potential. The ethanol oxidation pathway was proven to be affected in the knockout strain but not fully interrupted as suggested by partially intact co-metabolism with glucose and production of intermediate products.

Investigation of cofactor metabolism revealed that deletion of *mftG* resulted in a drastic increase in the ratio of reduced/oxidized mycofactocins in the mycobacterial cell while the NADH/NAD$^+$ ratio remained unchanged. This finding prompted the hypothesis that MftG could be involved in MFT cofactor regeneration. Assays with homologously and heterologously produced, semi-purified enzymes showed that MftG indeed oxidized PMFTH$_2$ and MMFT-2H$_2$, but only reactions reminiscent of a single-turnover process could be observed. Respirometry studies further demonstrated that reduced mycofactocins feed redox equivalents into the intact mycobacterial respiratory chain. Strikingly, this process was severely hampered in ΔmftG mutants, suggesting that MftG, while directly accepting electrons from mycofactocinol, delivers electrons either directly or indirectly into the respiratory chain. Transcriptomic analysis revealed a profound re-organization of the respiration machinery when MftG was absent suggestive of starvation and low availability of redox equivalents to the respiratory chain.

We, therefore, conclude that MftG is a mycofactocin dehydrogenase requiring electron acceptors associated with the membrane-bound respiratory chain. This assumption is plausible considering that other members of the GMC superfamily, for example choline dehydrogenase from *E. coli*, were reported to couple substrate oxidation with the electron transport chain (**Landfald and Strøm, 1986**). While the exact electron acceptor remains a subject for future studies, it is plausible that cytochromes might fulfill this function. Interestingly, cellobiose dehydrogenase, a fungal GMC enzyme, is a multi-domain enzyme where the dehydrogenase domain (GMC) is linked to a cytochrome domain (**Tan et al., 2015**). This situation is reminiscent of pyrroloquinoline quinone (PQQ)-dependent alcohol dehydrogenases (**Anthony, 2001**). Those enzymes, that mainly occur in the periplasm of Gram-negative bacteria, are typically coupled to a dedicated cytochrome c subunit, which is genetically clustered along with the other subunits (**Anthony, 2001**). It is also likely that the membrane-associated quinone pool serves as the electron acceptor. For instance, the GMC-family oxidoreductase enzymes PhcC and PhcD from *Sphingobium* sp. involved in lignin oxidation were suggested to transfer electrons to the ubiquinone pool (**Takahashi et al., 2015**). Reduction of menaquinone ($E_0' = -74$ mV), the dominant respiratory quinone of *Mycobacteria*, by mycofactocinol would be thermodynamically favorable considering the midpoint potential of premycofactocins (PMFT/PMFTH$_2$: $E_0' = -255$ mV; **Ayikpoe and Latham, 2019**). However, other redox coenzymes, redox proteins or protein complexes of the respiratory chain might act as direct reaction partners of MftG.

Our study sets the ground for future investigations into the connection of the mycofactocin system to further (known or unknown redox) systems. For instance, background MFTH$_2$ oxidation by membrane preparations of the ΔmftG mutant points towards the existence of alternative MFT-dependent dehydrogenases or oxidases. Our results provide insights that are important for mechanistic studies of previous effects shown to be related to MftG. For instance, previous studies reported growth defects connected to *mftG* inactivation by transposon mutagenesis in *M. tuberculosis* (**DeJesus et al., 2017**). More recently, the transcriptome analysis of a dose-dependent response of *M. tuberculosis* to the succinate dehydrogenase inhibitor BB2-50F showed that this new antibiotic induced upregulation of *mftG* (**Adolph, 2021**). Furthermore, vulnerability studies of *M. smegmatis*, *M tuberculosis* H37Rv and the hypervirulent strain HN878 using a CRISPRi library revealed that, although disruption of *mftG* did not induce any phenotype in *M. smegmatis*, still the disturbance of this gene induced growth defects in both *M. tuberculosis* strains (**Bosch et al., 2021**). Moreover, disruption of the *mftG* gene by this technique showed increased fitness of *M. tuberculosis* H37Rv when exposed to various drugs, for example bedaquiline and ethambutol, while reducing fitness when exposed to isoniazid and vancomycin (**Bosch et al., 2021**). Having these results in mind, further investigations of the role of MftG in *M. tuberculosis* survival and infection might disclose important implications of the MFT system for TB treatment.

Taken together, we close an important knowledge gap by showing that the flavoprotein MftG plays a role in MFT regeneration and acts as a redox shuttle between alcohol substrates and the respiratory chain. With the clarification of the MftG function, the entire set of *mft* genes are now elucidated in mycobacteria. Therefore, this study adds another important element to the amazingly flexible network

of mycobacterial redox metabolism and opens up an avenue for future investigations of the mycofactocin system.

## Materials and methods

### Bacterial strains and culture media

The *Mycolicibacterium smegmatis* MC$^2$ 155 strain was used in this study. All generated mycobacterial strains were maintained at 37 °C in lysogeny broth (LB) supplemented with 0.5 g L$^{-1}$ Tween 80 (LBT). *Escherichia coli* TOP10 (Thermo Fischer Scientific) was used to propagate plasmids on LB medium supplemented with either 100 µg mL$^{-1}$ hygromycin B (Carl Roth) or 50 µg mL$^{-1}$ kanamycin (Carl Roth). *E. coli* NiCo21(DE3) (New England Biolabs) was used to express recombinant proteins. Genomic manipulation of *M. smegmatis* was performed in LBT supplemented with 60 µg mL$^{-1}$ hygromycin B and 10 g L$^{-1}$ sucrose for double crossover event or 50 µg mL$^{-1}$ kanamycin for *gfp-hyg* cassette removal and complementation/overexpression strain generation. Studies for comparison between WT and mutants were performed with adapted Hartmans de Bont medium (*Ellerhorst et al., 2022*) (HdB-Tyl) containing (NH$_4$)$_2$SO$_4$ 2 g L$^{-1}$, MgCl$_2$·6H$_2$O 0.1 g L$^{-1}$, Na$_2$HPO$_4$ anhydrous 3 g L$^{-1}$, and KH$_2$PO$_4$ anhydrous 1.07 g L$^{-1}$ supplemented with 0.1% [v/v] 1000× trace element solution (ethylenediaminetetraacetic acid 10.0 g L$^{-1}$, CaCl$_2$·2H$_2$O 1.0 g L$^{-1}$, Na$_2$MoO$_4$·2H$_2$O 0.2 g L$^{-1}$, CoCl$_2$·6H$_2$O 0.4 g L$^{-1}$, MnCl$_2$·2H$_2$O 1.0 g L$^{-1}$, ZnSO$_4$·7H$_2$O 2.0 g L$^{-1}$, FeSO$_4$·7H$_2$O 5.0 g L$^{-1}$, and CuSO$_4$·5H$_2$O 0.2 g L$^{-1}$) and 0.5 g L$^{-1}$ tyloxapol (Sigma Aldrich) and supplemented with the appropriate carbon source: 10 g L$^{-1}$ absolute ethanol (TH.Geyer), 10 g L$^{-1}$ glucose (Carl Roth), 1% [v/v] glycerol (Carl Roth), 5 g L$^{-1}$ acetate (Carl Roth), 10 g L$^{-1}$ methanol (Carl Roth), 10 g L$^{-1}$ 1-propanol (Sigma-Aldrich), 5 g L$^{-1}$ 1-butanol (Sigma-Aldrich), 5 g L$^{-1}$ hexanol (Sigma-Aldrich), 0.010 g L$^{-1}$ acetaldehyde, 10 g L$^{-1}$ propane-1,2-diol (Sigma-Aldrich) and 10 g L$^{-1}$ propane-1,3-diol (Sigma-Aldrich).

### Bioinformatics analysis of MftG, co-occurrence and phylogenetic analysis

The AlphaFold (*Jumper et al., 2021*) structure prediction of MftG from *M. smegmatis* MC$^2$ 155 (model AF-I7F8I2-F1) was retrieved from the AlphaFold Protein Structure Database (*Varadi et al., 2022*). In order to investigate the FAD binding pocket, the closest homolog of MftG with a solved crystal structure, HMFO oxidase (PDB: 4UDP; *Dijkman et al., 2015*) was retrieved from the Protein Data Bank (PDB). The structure of HMFO oxidase was superimposed with the AlphaFold model of MftG using the matchmaker feature in ChimeraX version 1.2.5 (*Goddard et al., 2018*). The FAD molecule present in the HMFO oxidase crystal structure was used to complement the MftG model. The characteristic Rossman fold (GXGXXG) and histidine active site were identified through the alignment feature (MUSCLE) in Geneious Prime version 2022.2.2. comparing the aforementioned *M. smegmatis* protein sequence with examples of different activity GMCs previously described (*Aleksenko et al., 2020*). The regions were highlighted in the predicted structure.

The co-occurrence of *mftG* and *mftC* was analyzed similarly as described before (*Ellerhorst et al., 2022*). Since *mftC* is essential for the biosynthesis of mycofactocin, the presence of an *mftC* gene served as a proxy for the presence of the *mft* gene cluster. To obtain genomes encoding either an *mftC* or *mftG* homolog, the tables for the Hidden Markov Model (HMM) hits in the NCBI RefSeq protein database were retrieved from the respective NCBI protein family models' entries. The NCBI HMM accession numbers were TIGR03962.1 for *mftC* and TIGR03970.1, TIGR04542.1. and NF038210.1 for *mftG*. The MftG family was described by three distinct HMMs at the time of writing, with TIGR03970.1 as the main family comprising a variety of genera while the other two MftG models TIGR04542.1 and NF038210.1 describe homologs restricted to the genera *Gordonia* and *Dietzia*, respectively. The entries were filtered for origin in complete genomes and each entry was checked manually for the presence of potentially missed *mftC* or *mftG* homologs.

The search for other GMC proteins that could be present in the genomes of strains that contain the mycofactocin cluster (but no MftG) was performed by collecting a list of putative homologs of *M. smegmatis* MC$^2$ 155 Uniprot accession number A0QSC2 through BLASTP search of the non-redundant protein database from NCBI with the NCBI tool search with the following settings: max target sequences 5000, expect threshold 0.0002, word size 6, BLOSUM62, gap cost exist 11, gap cost extend 1, conditional compositional score matrix adjustment and filter low complexity regions

(*Altschul et al., 1990*). Partial sequences were not included in the analysis. The protein sequences of all GMC proteins obtained, as well as 16 previously published reference GMC enzymes, were used for phylogenetic analysis and are listed in *Supplementary file 2*, Table S1. The sequences were aligned using MAFFT (*Katoh et al., 2002*) and the phylogenetic tree was built using the maximum likelihood methods implemented in FastTree 2.1.11 (*Price et al., 2010*) with WAG substitution model (both programs implemented as Geneious plugins). The tree was further edited using TreeViewer Version 2.1 (*Bianchini and Sánchez-Baracaldo, 2023*) to condense phylogenetic clades and full tree was colored using FigTree v1.4.4.

## Plasmids and *M. smegmatis* genetic manipulation

The *M. smegmatis* MC$^2$ 155 mutants were produced using the pML2424/pML2714 (*Ofer et al., 2012*) system combined with the pMCpAINT complementation vector (*Krishnamoorthy et al., 2019*). The plasmid for *mftG* removal (pPG17) resulted from the combination of the vector pML2424 with SpeI/SwaI and PacI/NsiI ligation with the amplified upstream and downstream regions of *mftG* using primers described in *Table 1*, flanking the *gfp-hyg* cassette. Preparation of mycobacterial competent cells and transformation was conducted as previously described (*Ellerhorst et al., 2022*). The confirmation of the mutant (Δ*mftG*) was achieved with PCR amplification of external and internal primers (*Table 1*) after the removal of the selection cassette using Cre recombinase from pML2714. The Δ*mftG* was made competent for complementation (Δ*mftG-mftG*) and overexpression (WT-*mftG*) of *mftG* by genome integration using plasmid pPG29, which originated from pMCpAINT as a backbone by the addition of a NdeI restriction site using the primers listed in *Table 1*. The linearized plasmid was cut with NdeI and BamHI for the introduction of the native *M. smegmatis* MC$^2$ 155 *mftG* gene (synthesized by BioCat). To obtain the overexpression strain (WT-*mftG*), the WT strain was transformed with pPG29. The plasmid pPG32 was obtained by PCR using pPG29 as a template and appropriate primers to add the hexa-histidine tag at the C-terminus (*Table 1*). It was subsequently used to transform the Δ*mftG* strain and used for *mftGhis₆* expression (Δ*mftG-mftG*His$_6$). The plasmid pPG23 was constructed using pPG20 as a backbone, introducing at the NcoI/HindIII cutting sites the PCR-amplified mycofactocin operon *mftA-F* (primers listed in *Table 1*) and used to generate the strain WT-*mftABCDEF* with the integration of the plasmid in the WT strain. The PCR amplifications were performed using Q5 High-Fidelity DNA Polymerase (New England Biolabs) with supplementation with High GC-enhancer to the reaction following the manufacturer's instructions and primers synthesized by Eurofins Genomics. The plasmids were constructed using T4 DNA ligase (New England Biolabs) with a molar ratio of 3:1 of insert to backbone. All the molecular Biology work was planned using Geneious Prime version 2022.2.2. (https://www.geneious.com/). All plasmids and strains are listed in *Table 1*, plasmid sequences are available in the **Supplementary Information** Text 1.

## Growth curves of *M. smegmatis*

Growth curves of *M. smegmatis* WT, Δ*mftG*, Δ*mftG-mftG,* and WT-*mftG* using HdB-Tyl supplemented with 10 g L$^{-1}$ ethanol or 10 g L$^{-1}$ glucose as the sole carbon source at 37 °C and 210 rpm were performed three times in biological triplicates. Evaluation of growth on 1% [v/v] glycerol, 5 g L$^{-1}$ acetate, 10 g L$^{-1}$ methanol, 10 g L$^{-1}$ 1-propanol, 5 g L$^{-1}$ 1-butanol, 0.5% hexanol [v/v], 0.010 g L$^{-1}$ acetaldehyde, 10 g L$^{-1}$ propane-1,2-diol, 10 g L$^{-1}$ propane-1,3-diol were performed in at least duplicates. Culture of the strains grown on HdB-Tyl supplemented with 10 g L$^{-1}$ glucose at 37 °C and 210 rpm for 24 h was used as pre-inoculum. Cultures were centrifuged at 4000×*g* for 5 min and the supernatant was removed. The pellet was resuspended with base HdB-Tyl and used as inoculum for new cultures in a total volume of 40 mL in 250 mL Erlenmeyer flasks (50 mm Ø) with starting OD$_{600}$ of 0.1 and closed with breathable rayon film (VWR). The cultures were monitored using the Cell Growth Quantifier (Aquila Biolabs) with readings every 60 s and incubated at 37 °C and 210 rpm until the late stationary phase was reached. Recorded data were further processed using CGQquant (Aquila Biolabs) for merging the replicates. The resulting average and standard deviation data were plotted in GraphPad Prism 9.

## Ethanol, acetaldehyde, and acetate quantification

Cultures of the WT, Δ*mftG*, Δ*mftG-mftG,* and WT-*mftG* were grown (starting OD$_{600}$ of 0.1, 37 °C, 210 rpm) in triplicates in HdB-Tyl supplemented with 10 g L$^{-1}$ ethanol for 24 h, 32 h, 48 h, and 72 h and quantified for ethanol, acetaldehyde, and acetate concentration. The culture of Δ*mftG* starting

**Table 1.** List of *M. smegmatis* and *E. coli* strains, vectors, plasmids and primers used and generated on the course of this study.

| Strain | Description | Reference |
|---|---|---|
| WT | *Mycolicibacterium smegmatis* MC$^2$ 155 | ***Krishnamoorthy et al., 2019*** |
| Δ*mftG* | derivate of WT without *mftG* replaced with *loxP* site | this study |
| Δ*mftG-mftG* | derivate of Δ*mftG* integrated with pPG29 at the *attB* site | this study |
| WT-*mftG* | derivate of WT integrated with pPG29 at the *attB* site | this study |
| Δ*mftG- mftG*His$_6$ | derivate of Δ*mftG* integrated with pPG32 at the *attB* site | this study |
| WT-*mftABCDEF* | derivate of WT integrated with pPG23 at the *attB* site | this study |
| NiCo21(DE3) Competent *E. coli* | derived from *E. coli* BL21 (DE3) | New England Biolabs |
| *Escherichia coli* TOP10 | F$^-$*mcr*A Δ(*mrr-hsd*RMS-*mcr*BC) φ80*lacZ*ΔM15 Δ*lac*X74 *rec*A1 *ara*D139 Δ(*ara-leu*)7697 *gal*U *gal*K  λ$^-$*rps*L(Str$^R$) *end*A1 *nup*G | Thermo Fischer |

| Plasmid name | Backbone | Reference |
|---|---|---|
| pML2424 | vector for double crossover event with tdTomato, *gfp-hyg* cassette, and PAL5000ts | ***Ofer et al., 2012*** |
| pML2714 | vector with kanamycin resistance for Cre recombinase expression and *gfp-hyg* cassette removal | ***Ofer et al., 2012*** |
| pPG20 | pMCpAINT derivate with kanamycin resistance, potential mycofactocin promotor, and *mftF* | ***Peña-Ortiz et al., 2020a*** |
| pPG17 | pML2424 with up and downstream regions of *mftG* | this study |
| pPG23 | pMCpAINT derivate with kanamycin resistance, potential mycofactocin promotor, and *mftABCDEF* | this study |
| pPG29 | pPG20 with *mftF* replaced with *mftG* | this study |
| pPG32 | pPG29 with *mftG* replaced with *mftG*His$_6$ | this study |
| pPG36 | pMAL-C4X with *malE* fused with *mftG* codon optimized for *E. coli* expression | this study |

| Primer name | Primer sequence 5'–3' | Amplicon |
|---|---|---|
| GMC_up_F1 | GCTACACTAGTCGGTGTCGTATGTGCCGAG | upstream region of *mftG* |
| GMC_up_R1 | GCTACATTTAAATTCAAAGTCGGCGGCTAACTC | |
| GMC_dn_F1 | GCTACTTAATTAATCGACGGCTCGATCATGC | downstream region of *mftG* |
| GMC_dn_R1 | GCTACATGCATGTTGTCGAGGCTCCGGTG | |
| INT_GMC_F1 | CACTATGGGTCGACGCTGAC | internal region of *mftG* |
| INT_GMC_R1 | GCGTGACTTACCAATTCGCG | |
| EXT_GMC_F1 | AACATCGTGGCCCGGTAC | external region of *mftG* |
| EXT_GMC_R1 | CTCCTCACGCGACGACTC | |
| pMCpAINT_FC_F | GCTACAAGCTTATCGATGTCGACGTAGTTAAC | backbone pMCpAINT introducing NdeI |
| pPG20_NdeI_R | GCTACCCATATGCGTATGGTCTCGACAGTTGT | |
| GMC_COMP_F1 | GCTACCCATATGGAGTTAGCCGCCGACTTT | insertion of 6 histidines C-terminally |
| GMC_Hist_R3 | GCTACAAGCTTACTATTAGTGGTGGTGGTGGTGGTGGGTCGCGATG AACTCGGC | |
| pMCpAINT_conf_2_F | CTGATACCGCTCGCCGCA | sequencing confirmation |
| pMCpAINT_conf_2_R | CTTTCGACTGAGCCTTTCGT | |
| MFTKIMS_FC_KI_CLUSTER_F2 | GCTACCCATGGTCGGACATCTCTCACACCCC | region from hypothetical mycofactocin precursor until end of *mftF* |
| MS_FC_KI_CLUSTER_R1 | GTTAACTACGTCGACATCGATAAGCTTTCAAAGTCGGCGGCTAACTC | |

at $OD_{600}$=1 and medium without inoculum were also analyzed. The samples were centrifuged at 17,000×$g$ for 20 min and supernatants were sterilized using 0.2 µm cellulose acetate membrane filters (VWR) and kept at –20 °C until further analysis. The samples were diluted 1:10 with 0.005 mol $L^{-1}$ $H_2SO_4$ and 50 µL injected in an HPLC X-LC (JASCO International Co) using a pre-column Kromasil 100 C18, 40 mm x 4 mm, 5 µm (Dr. Maisch GmbH, Ammerbuch-Entringen) combined with the column Aminex HPX-87H Ion Exclusion Column, 300 mm x 7.8 mm, 9 µm (Bio-Rad) with isocratic mobile phase 0.005 mol $L^{-1}$ $H_2SO_4$, flow 0.5 mL $min^{-1}$, heated to 50 °C. Detection was performed via refractive index (RI) and UV (210 nm) and compared with standards of ethanol (Uvasol for spectroscopy, Merck), acetaldehyde (Reagent plus, Sigma-Aldrich), and acetic acid (100%, water-free, p.a., Merck). Acetaldehyde content was further quantified using the supernatant of WT and Δ$mftG$ grown in HdB-Tyl supplemented with 10 g $L^{-1}$ ethanol alone and 10 g $L^{-1}$ glucose combined with 10 g $L^{-1}$ ethanol for 48 h as described in the Acetaldehyde Assay Kit (Sigma-Aldrich) by the manufacturers.

## Flow cytometry measurements

The cultures of WT, Δ$mftG$, Δ$mftG$-$mftG$, and WT-$mftG$ grown in HdB-Tyl supplemented with 10 g $L^{-1}$ glucose, 10 g $L^{-1}$ ethanol, 10 g $L^{-1}$ glucose and 10 g $L^{-1}$ ethanol combined or no carbon source (starvation) incubated at 37 °C and 210 rpm were sampled at 24 hr, 48 hr, and 72 hr, centrifuged at 4000×$g$ for 5 min and the supernatant removed. Samples were re-suspended in 500 µL HdB-Tyl with no carbon source with $OD_{600}$ adjusted to 0.2. To samples of each condition were added either 2.5 µL of 750 µM of propidium iodide; 3 µL of 3 mM 3,3'-diethyloxacarbocyanine iodide [$DIOC_2$(3)]; 20 µL of 500 mM of the protonophore (uncoupler) carbonyl cyanide 3-chlorophenylhydrazone (CCCP, Sigma-Aldrich) followed by 3 µL of 3 mM $DIOC_2$(3) combined. Samples were incubated for 15 min and briefly vortexed. All the samples were injected on a FACS AriaFusion (BD Biosciences) and for each sample, FSC and SSC were detected using the blue laser (488 nm) and a threshold set to 400. The propidium iodide is excited by 488 nm and has an emission of 630 nm and emission was detected with the help of a 600 nm long-pass filter and a 610/20 nm bandpass filter. $DiOC_2$(3) has the excitation maximum at 488 nm and the cells with a low level of transmembrane potential (or the cells treated with uncoupler CCCP) have a maximum emission at 530 nm (green color), the cells with a higher level of electric potential energy accumulate more concentration of this lipophilic dye, which results in its accumulation within the cells and these aggregates of dye have a maximum of emission at 600 nm (red fluorescence; *Novo et al., 1999*; *Nikitushkin et al., 2020*). The emission of $DIOC_2$(3) was detected with the help of a 502 nm long-pass filter and a 530/30 nm bandpass filter and 600 nm long-pass filter and a 610/20 nm bandpass. PMT voltages were adjusted to values of 600 V for the red channel and 350 V for the blue channel. The propidium iodide data were analyzed in a batch using flowCore (*Hahne et al., 2009*) and ggcyto (*Van et al., 2018*) R packages in R (version 4.1.0). Single cells were gated using FSC.A vs. FSC.H (intensities below 0.5e3 and above 1e5 have been discarded) and in the following step the highly fluorescent single cells were gated using YG610.A vs. FSC.A. The transmembrane potential data ($DIOC_2$(3) and CCCP) were directly gated using FlowJo v10.8 Software (BD Life Sciences) in coordinates B530.A~Y610.A (*Nikitushkin et al., 2020*).

## Phenotype microarrays

The full panel of phenotype microarrays (PMs) on BioLog plates was used to detect additional phenotypes. The initial cultures of *M. smegmatis* WT and Δ$mftG$ were grown in LB overnight at 37 °C and 210 rpm, after 24 h the cultures were upscaled to 50 mL in HdB-Tyl supplemented with 10 g $L^{-1}$ glucose. Preparation of *M. smegmatis* cells for PMs was performed with minor changes compared to the one previously described (*Karlikowska et al., 2021*). After 24 hr, the cultures were centrifuged at 4000×$g$ for 5 min and the supernatant was removed. The cells were resuspended in base HdB-Tyl and stored for 22 h at room temperature for starvation. The cultures were once again centrifuged at 4000×$g$ for 5 min, the supernatant discarded and the cells resuspended in GN/IF-0 to an $OD_{600}$ of 0.68. The culture was then supplemented with a PM mixture. The culture mixture of the WT and Δ$mftG$ was inoculated with 100 µL in each well of the 20 plates and incubated at 37 °C for 48 hr. The $OD_{595}$ was measured in a CLARIOStar microplate reader (BMG Labtech) after 48 h and 72 h. The sensitivity plates data was analyzed towards differences of growth compared to control growth of each strain and between Δ$mftG$ and WT. Only differences in growth percentages comparing WT and

$\Delta mftG$ [(%$\Delta mftG$ *100)/ % WT] above 150% or under 50% were considered for discussion and results were plotted using GraphPad Prism 9.

## Fluorescent D-amino acid (FDAA) labeling

To target peptidoglycan biosynthesis, fluorescently labeled D-amino acids were incorporated into the nascent peptidoglycan as previously reported (*Kuru et al., 2019*). Briefly, WT, $\Delta mftG$, and $\Delta mftG$-*mftG* cells were grown in HdB-Tyl supplemented with either 10 g L$^{-1}$ glucose, 10 g L$^{-1}$ ethanol, or 10 g L$^{-1}$ glucose combined with 10 g L$^{-1}$ ethanol. The starvation condition was accomplished with the incubation of bacteria in a plain HdB-Tyl medium with no carbon source added. The cultures were incubated at 37 °C and 210 rpm until the mid-logarithmic growth phase. For FDAA incorporation, the green NADA was added to a final concentration of 250 µM, and cells were left to grow for 2.5 hr. To stop incorporation, cells were placed on ice for 2 min followed by one washing step with ice-cold PBS. The red RADA was added to the same final concentration for 2.5 hr. After the final washing step with PBS, bacterial cells were fixed with 4% (v/v) paraformaldehyde (PFA) solution for 20 min at 4 °C.

## Super-resolved structured illumination microscopy (SR-SIM)

For the SR-SIM imaging, 10 µL of the sample was spotted on 10 g L$^{-1}$ agarose pads. The agarose pads were covered with 1.5 H coverslips (Roth) and stored at 4 °C for further imaging. The SR-SIM data were acquired on an Elyra 7 system (Zeiss) equipped with a 63×/1.4 NA Plan-Apochromat oil-immersion DIC M27 objective lens (Zeiss), a Piezo stage, and a PCO edge sCMOS camera with 82% QE and a liquid cooling system with 16-bit dynamic range. Using Lattice SIM mode, images were acquired with 13 phases. NADA was detected with a 488 nm laser and a BP 495–590 emission filter; RADA was detected with a 561 laser and an LP 570 emission filter. Super-resolution images were computationally reconstructed from the raw data sets using default settings on ZenBlack software (Zeiss). Images were analyzed using the Fiji ImageJ software (*Schindelin et al., 2012*).

## NADH/NAD$^+$ and ADP/ATP ratio measurement

The *M. smegmatis* WT and $\Delta mftG$ strains were grown in triplicates on HdB-Tyl supplemented with 10 g L$^{-1}$ ethanol as the sole carbon source and incubated at 37 °C and 210 rpm with starting OD$_{600}$=0.1 for WT and OD$_{600}$=1 for $\Delta mftG$ for a period of 48 hr. Samples of culture were adjusted to OD$_{600}$ 0.5 mL$^{-1}$ and processed according to the manufacturer's instruction ADP/ATP Ratio Assay Kit (Sigma-Aldrich). For NADH/NAD$^+$ quantification samples from WT and $\Delta mftG$ grown in triplicates on HdB-Tyl supplemented with 10 g L$^{-1}$ glucose for 24 h were also quantified. Different cell concentrations were tested with the best result OD$_{600}$ of 0.1. Bacterial suspensions were diluted in ice-cold PBS (8 g L$^{-1}$ NaCl, 0.2 g L$^{-1}$ KH$_2$PO$_4$, 1.15 g L$^{-1}$ Na$_2$HPO$_4$, 0.2 g L$^{-1}$ KCl, at pH 7.4), samples were centrifuged at 10,000 x $g$, 4 °C, for 5 min, and the resulting pellets were extracted with 100 µL of extraction buffer either for NAD $+$or NADH extraction. The resulting extracts were treated and measured according to the NAD$^+$/NADH Assay Kit manufactures' protocol (MAK460, Sigma-Aldrich). The readings of each plate were acquired using a CLARIOStar microplate reader (BMG Labtech) and data were further plotted and statistical analysis performed in GraphPad Prism9.

## MFT profiling

The workflow for metabolome extraction was based on our previously described protocol with small changes (*Ellerhorst et al., 2022*). The *M. smegmatis* WT and $\Delta mftG$, cultures were incubated in HdB-Tyl with either 10 g L$^{-1}$ glucose or 10 g L$^{-1}$ ethanol at 37 °C and 210 rpm in triplicates. Cultures were quenched at the exponential phase, for WT and $\Delta mftG$ strain at 30 h in glucose and WT strain at 35 h in ethanol. The $\Delta mftG$ strain grown in ethanol was quenched at 60 h when it presented a small growth. Cultures were also sampled at stationary phases, for WT and $\Delta mftG$ strains at 45 h in glucose and WT strain at 60 h in ethanol. The *M. smegmatis* $\Delta mftG$-*mftG* and WT-*mftG* strains were grown in HdB-Tyl with 10 g L$^{-1}$ ethanol at 37 °C and 210 rpm in triplicates and quenched only at the stationary phase (60 hr). The samples were normalized upon sampling to 10 mL of 1 unit of OD$_{600}$, quenched in 20 mL of cold extraction mix (acetonitrile:methanol:water: FA; 60:20:19:1, v/v), and extracted as described previously (*Peña-Ortiz et al., 2020a*). The lyophilized samples were dissolved with 450 µL LC-MS grade water (VWR) twice, combined, and the extracts were centrifuged at 17,000×$g$ for 20 min twice to remove debris and kept at –20 °C until analysis. The metabolomes were analyzed using 10 µL

injection in LC-MS/MS, Dionex UltiMate 3000 UHPLC connected to a Q Exactive Plus mass spectrometer (Thermo Fisher Scientific), using the previously described method (**Peña-Ortiz et al., 2020a**).

LC-MS/MS raw files were loaded into MZmine version 2.53 (**Pluskal et al., 2010**) and processed via the following pipeline for targeted peak analysis. Mass lists were created on MS[1] and MS[2] levels using the mass detector module set to centroid with a noise level of 0. Mass detection was followed by targeted peak integration using the targeted feature detection module set to MS level: 1, intensity tolerance: 50%, noise level: 0, $m/z$ tolerance: 0 $m/z$ or 5 ppm, retention time tolerance: 0.2 min, and a target list comprising expected protonated ions of the most abundant mycofactocin congener found in vivo: Methylmycofactocinol-8 (MMFT-8H$_2$, sum formula: $C_{62}H_{99}NO_{43}$, expected retention time: 6.81 min, expected $m/z$ 1546.5663 [M+H]$^+$) and the corresponding oxidized form Methylmycofactocinone-8 (MMFT-8, sum formula: $C_{62}H_{97}NO_{43}$, expected retention time: 7.18 min, expected $m/z$ 1544.5507 [M+H]$^+$). Theoretical mass-to-charge ratios were calculated using the respective sum formulae and the enviPat web interface (**Loos et al., 2015**). Resulting feature lists were filtered for the presence of at least one entry to remove empty rows and aligned using the RANSAC aligner module with the following settings: $m/z$ tolerance: 0 $m/z$ or 5 ppm, retention time tolerance: 0.2 min before and after correction, RANSAC iterations: $10^4$, the minimum number of points: 20%, threshold value: 1, linear model: false, require same charge state: false. The resulting aligned feature list was manually inspected for misidentifications and exported as a comma-separated value table. Statistical analysis and plotting were performed with GraphPad Prism 9. Ratios between oxidized and reduced forms were calculated using MMFT-8H$_2$/MMFT-8.

## Purification of methylmycofactocinol-2 (MMFT-2H$_2$) from WT-*mftABCDEF*

To isolate MMFT-2H$_2$ from *M. smegmatis*, the mycofactocin gene cluster duplication strain WT-*mftABCDEF* was built by integration of the plasmid pPG23 in the WT strain. A first preculture was prepared through inoculation of 3 mL LBT with the strain and incubation for 24 hr. The first preculture was used to inoculate a second preculture of 50 mL HdB-Tyl supplemented with 10 g L$^{-1}$ glucose in a 250 mL shake flask and cultivated for 48 hr. From the second preculture 1:100 [v/v] was used to inoculate the 5 L main culture of 2 x HdB, that is all components except tyloxapol were concentrated twice compared to standard HdB-Tyl, supplemented with 3% [v/v] ethanol. The main culture was cultivated as 10x500 mL cultures in 2 L shake flasks sealed with breathable rayon film (VWR) for reproducible aeration for 72 hr. All cultivation steps were performed at 37 °C and 210 rpm in a shaker with a 25 mm throw. The culture was harvested through centrifugation at 8,000×$g$ and 18 °C for 3 min. The cell pellet of the harvested culture was washed with 500 mL ultrapure water once followed by centrifugation at 8000×$g$ and 18 °C for 5 min and resuspension in 500 mL of 50 mM citrate phosphate buffer (Na$_2$HPO$_4$ anhydrous 7.1 g L$^{-1}$, citric acid 9.6 g L$^{-1}$, pH 5.2). The cell suspension was autoclaved at 134 °C for 30 min to disrupt the cells followed by treatment with 20 µg mL$^{-1}$ of cellulase mix from *Trichoderma reesei* (Sigma Aldrich) at 37 °C for 18 h, which was subsequently heat-inactivated at 95 °C for 15 min to condense the pool of different long-chain mycofactocinols present in vivo to MMFT-2H$_2$ and PMFTH$_2$. The treated crude extract was cleared through centrifugation at 15,000×$g$ and 18 °C for 20 min. To avoid column overloading, the extract was split into two 250 mL parts before fractionation on 70 mL/10 g octadecyl modified silica solid phase extraction columns (Chromabond SPE, Machery-Nagel). Before fractionation, the SPE column was washed with three column volumes of methanol (CV, 1 CV = 40 mL) followed by equilibration with three CVs of 50 mM citrate phosphate buffer. After application of the crude extract, bound molecules were washed with 1 CV water and step-wise eluted with 4 CV 80:20 [v/v] water:methanol, 3 CV 70:30 [v/v] water:methanol, and 1 CV 50:50 [v/v] water:methanol. Samples of each fraction were tested for the presence of MMFT-2H$_2$ via LC-MS as described above. The 70:30 fraction showed the highest concentration of MMFT-2H$_2$ and thus was subjected to further purification. The fraction was lyophilized at 1 mbar and −90 °C until all solvent was removed. The lyophilizate was resuspended in 500 µL water and subjected to a final purification via size-exclusion chromatography (SEC) using a Superdex 30 Increase 10/300 GL (Cytiva) equilibrated with 50 mL water operated at 500 µL min$^{-1}$ on an NGC medium-pressure chromatography system (Bio-Rad). Per chromatography, 250 µL resuspended enriched fractions were separated. Each of the SEC fractions was tested for the presence of MMFT-2H$_2$ via LC-MS/MS as described above and the fractions with the highest concentrations of

MMFT-$2H_2$ and lowest amounts of LC-MS/MS detectable impurities were combined and lyophilized at 0.01 mbar and –90 °C until complete dryness.

## Synthesis of PMFT and PMFTH$_2$

Premycofactocinone (PMFT) and premycofactocinol (PMFTH$_2$) were synthesized as reported previously (*Ellerhorst et al., 2022*).

## Homologous MftG expression and assay

The strain Δ*mftG*-*mftG*His$_6$ was designed for the homologous expression of a hexahistidine-tagged MftG protein. In order to confirm the correct expression and activity of the induced protein the Δ*mftG*-*mftG*His$_6$ was grown in HdB-Tyl supplemented with 10 g L$^{-1}$ ethanol to confirm recovery from the phenotype. The strains Δ*mftG* and Δ*mftG*- *mftG*His$_6$ were inoculated in 200 ml HdB-Tyl supplemented with 5 g L$^{-1}$ glucose and 20 g L$^{-1}$ ethanol in 1 L flasks covered with breathable membrane and starting OD$_{600}$ of 0.1. The cultures were incubated for 48 h at 37 °C and 210 rpm and the cell pellet were retrieved after centrifugation at 8000 rpm for 20 min at 4 °C and stored at –20 °C. The cell pellet of both strains was resuspended in 8 mL PBS buffer with 5 mM imidazole, 300 mM NaCl, 50 µL protease inhibitor (1/100 solution), and 10 µM flavin adenine dinucleotide (FAD). The lysates were retrieved by sonicating (Bandelin Sonoplus UW 2070 homogenizer) the cells for 7.5 min (30 s on/off) at 70% cycle time and 70% amplitude in cold conditions and collection of supernatants after centrifugation at 15,000 rpm for 30 min at 4 °C. Lysates were incubated for 1 h 30 min with 2 mL Ni-NTA beads on ice under constant shaking. The beads and lysate were transferred to an empty column and flow-through discarded, the column was washed with 5 mM (2 CV) and 20 mM (2 CV) imidazole by gravity flow. The partially purified fraction was eluted using 100 mM imidazole and the buffer was changed to 0.1 M sodium phosphate at pH 7 using a PD-10 column. The proteins were concentrated using a Vivaspin6 10 kDa filter unit (Sartorius). The protein concentration was measured via absorption at 280 nm using a Nanodrop spectrophotometer (ThermoFisher) and adjusted to 0.500 mg mL$^{-1}$.

From each extract, 20 µL were assayed with 20 µL of Δ*mftG* metabolite extract and incubated at 37 °C, an initial sample was quenched with acetonitrile (1:1) directly after the mixture and final samples were quenched after 18 h of incubation. Pure substrates at 2 µM of MMFT-$2H_2$, synthetic PMFTH$_2$, and PMFT (*Ellerhorst et al., 2022*) were also used in the enzymatic assays, and samples every 20 min for a total of 2 h were retrieved and quenched as before. Heat-inactivated cell-free extract was used to confirm enzyme-dependent activity. All variables were performed in triplicates. The extracts were centrifuged at 17,000×*g* for 20 min and the supernatant was retrieved and centrifuged a second time in the same conditions to remove debris. From each assay, 10 µL were injected in LC-MS/MS and analyzed as described above. Differences of MMFT-$8H_2$ and MMFT-8, MMFT-$2H_2$ and MMFT-2, PMFTH$_2$ and PMFT amounts between initial and different incubation points were retrieved with targeted analysis to the later compounds using MZmine version 2.53 (*Pluskal et al., 2010*).

## Heterologous MftG expression and activity

*Escherichia coli* NiCo21(DE3) was transformed with pPG36 as described. The pMAL-c4x-based plasmid harbored a gene for the expression of a maltose-binding protein fused N-terminally to MftG from *M. smegmatis* (WP_014877070.1) codon-optimized for the expression in *E. coli* obtained from BioCat. A single colony of *E. coli* NiCo21(DE3)/pPG36 was used to inoculate a preculture on 5 mL LB which was incubated at 37 °C and 210 rpm for 18 hr. Three mL of the preculture were used to inoculate 300 mL of LB in a 1 L shake flask. The expression culture was cultivated at 37 °C and 210 rpm until an OD$_{600}$ of 0.6, when gene expression was induced through the addition of isopropyl β-D-1-thiogalactopyranoside (IPTG) at a final concentration of 1 mM. The induced expression culture was incubated at 18 °C and 210 rpm for 20 h. Media were supplemented with 100 µg mL$^{-1}$ of ampicillin and cultivation was performed using an incubator with a throw of 25 mm. Cells were harvested through centrifugation at 8000 x *g* and 4 °C for 3 min. The cell pellet was resuspended in 12 mL lysis buffer (20 mM Tris, 200 mM NaCl, 1 mM EDTA, pH 7.4) and disrupted through sonication using a Bandelin Sonoplus UW 2070 homogenizer equipped with an MS 73 microtip operated at 70% cycle time and 70% amplitude for a total on-time of 7 min with cycles of 1 min sonication followed by 30 s pause to cool the lysate. Sonication was performed on ice. The lysate was cleared through centrifugation at 20,000 x *g* and 4 °C for 30 min followed by filtration through a 0.22 µm cellulose-acetate filter. The

cleared lysate was subjected to affinity chromatography using a 1 mL MBPTrap HP maltose-binding protein affinity chromatography column (Cytiva) equilibrated with 10 mL lysis buffer. Bound proteins were washed with 20 mL of lysis buffer and eluted using 10 mL of elution buffer (20 mM Tris, 200 mM NaCl, 1 mM EDTA, 10 mM Maltose, pH 7.4). Chromatography was performed with cooled buffers and at a flowrate of 1 mL min$^{-1}$. Target protein-containing fractions were combined and concentrated using an Amicon Ultra-4 50 kDa filter (Merck) operated at 7500 x $g$ and 4 °C. Protein concentrations were determined using the A$_{280}$ nm method of a NanoDrop (Thermo Fisher Scientific) with the calculation setting of 1 ABS = 1 mg mL$^{-1}$. Mycofactocinol oxidation assays were performed in a total reaction volume of 30 µL in lysis buffer including 2 µM MMFT-2H$_2$, 100 µM FAD, and either one of the tested potential electron acceptors: 2,6-dichlorophenolindophenol (DCPIP), nicotinamide adenine dinucleotide (NAD), N,N-dimethyl-4-nitrosoaniline (NDMA), phenazine methosulfate (PMS), at a concentration of 100 µM and 87 µg protein, or no artificial electron acceptor and at different protein concentrations as indicated (*Figure 6E*). The reactions were incubated at 37 °C for 24 h and quenched through heat-inactivation of the enzyme at 100 °C for 10 min. The quenched reactions were centrifuged at 17,000 x $g$ and 18 °C for 20 min and 10 µL of the supernatant were subjected to LC-MS/MS analysis with subsequent targeted data analysis as described. The target features for data analysis were the mycofactocin substrate and expected product of the assay, namely methylmycofactocinol-2 (MMFT-2H$_2$, sum formula: C$_{26}$H$_{39}$NO$_{13}$, expected retention time: 7.10 min, expected $m/z$ 574.2494 [M+H]$^+$) and methylmycofactocinone-2 (MMFT-2, sum formula: C$_{26}$H$_{37}$NO$_{13}$, expected retention time: 7.47 min, expected $m/z$ 572.2338 [M+H]$^+$). All experiments were performed in triplicates and controls were set up with lysis buffer replacing the enzyme. All tested potential electron acceptors were purchased from Sigma-Aldrich.

## Whole-cell respiration assay

The WT and Δ*mftG* cells were incubated on HdB-Tyl supplemented with 10 g L$^{-1}$ ethanol for 48 h at starting OD$_{600}$ of 0.1 for WT and 0.7 for Δ*mftG* at 37 °C and 210 rpm. For the starvation experiment, WT was cultivated for 24 h in the same conditions as the latter in HdB-Tyl without carbon sources and starting OD$_{600}$ of 0.7. After this period, the cells were collected by centrifugation at 5000×$g$ for 10 min and pellets were resuspended in fresh HdB supplemented with 0.25 g L$^{-1}$ Tyloxapol without carbon sources and normalized to OD$_{600}$ of 0.7.

Aliquots of 200 µL of bacterial samples were placed in the electrochemical chamber supplied with a Clark-type polarographic sensor (Oxytherm$^+$, Hansatech Instruments). Before experiments, the chamber was calibrated according to the manufacturer's manual ('liquid phase calibration') and set to 37 °C and 50 rpm stirring. The oxygen consumption was monitored over 10 min. The retrieved data was further plotted using R version 2023.03.0+.

## Isolation of mycobacterial membranes

*M. smegmatis* WT and Δ*mftG* strains were grown in 320 mL HdB-Tyl medium supplemented with 1 g L$^{-1}$ ethanol as the sole carbon source in 2 L flasks covered by breathable rayon film (VWR) in triplicates and incubated at 37 °C and 210 rpm with a starting OD$_{600}$ of 0.1 for WT and 2.0 for Δ*mftG* for a period of 48 hr.

The cells were concentrated by centrifugation at 7000×$g$ for 10 min and the resulting pellets were resuspended in buffer (38 mL) of the following composition: 50 mM Tris-HCl, 300 mM sucrose, 5 mM EDTA, 50 mM MgCl$_2$ at pH 6.7. Pellets were disrupted with 0.1 mm ZrO$_2$ beads (Carl Roth, GmbH) in 15 mL plastic tubes using a bead beater (MP Biomedicals, FastPrep-24 5 G). A total of 6 cycles of beating were applied (3 times of application of the pre-installed program for *M. tuberculosis* cells from the manufacturer). After each step of beating, tubes were cooled down on ice for at least one minute. To get rid of the beads and undisrupted cells the resulting material was centrifuged twice at 4 °C and 6000×$g$ for 15 min. The obtained lysate was further ultra-centrifuged (Thermo Scientific Sorvall LYNX 6000 Superspeed Centrifuge) at 4 °C, 100,000×$g$ for 2 h to obtain the sedimentation of the mycobacterial membrane fraction and clarified soluble fraction (cytoplasm and disrupted periplasm). The isolated membranes were stored on ice for no longer than 12 hr.

## Analysis of MMFT-2H$_2$ and PMFTH$_2$ oxidation in respiration assay

Oxygen consumption was measured using Oxytherm$^+$ (Hansatech Instruments) advanced oxygen electrode system at 37 °C and stirred at 50 rpm. For homogenization of mycobacterial membranes, the assay buffer of the following composition was used: 50 mM sucrose, 5 mM MgCl$_2$, 5 mM KF, 2 mM AMP, 4 mM K$_2$HPO$_4$, 50 mM Tris at pH 6.7. All experiments were performed with 250 µL of membrane homogenates. The respiration stimulation started after circa 100 s from the beginning of the oxygen measurement experiments by the addition of 1 mM of the substrates: MMFT-2H$_2$, PMFTH$_2$, NADH (Carl Roth), or sodium succinate (Sigma-Aldrich). Inhibition of the reaction was tested by the addition of a final concentration of 2.5 mM KCN (Sigma-Aldrich) to the solution. To exclude non-enzymatic oxidation, controls such as reaction mixtures without substrates, as well as microwave-inactivated membrane fractions were included. Total protein concentration was quantified using Roti-Nanoquant 5 x Concentrate (Carl Roth) and normalization on the total protein concentration of the fractions was used to make the results comparable. The respirometry curves were plotted using R version 2023.03.0+.

For the analysis of PMFTH$_2$ conversion into PMFT, 200 µL of the sample from the respirometry analysis was quenched by 200 µL of acetonitrile followed by two centrifugations at 17,000×$g$ for 15 min to remove debris. To assess if PMFTH$_2$ conversion was membrane-bound, an assay was carried out with soluble fractions and further quenched as previously described. The LC-MS/MS profiling and the obtained chromatograms were analyzed as described above in this study. Generated csv tables were further plotted and analyzed in R version 2023.03.0+.

## RNA extraction, library preparation and analysis of *M. smegmatis* MC$^2$ 155 WT and Δ*mftG*

*M. smegmatis* MC$^2$ 155 WT and Δ*mftG* were grown on HdB-Tyl with 10 g L$^{-1}$ for 24 hr. Cultures were centrifuged, supernatant removed and resuspended on HdB-Tyl plain medium and used as inoculum for triplicate cultures on HdB-Tyl with 10 g L$^{-1}$ of ethanol at a starting OD$_{600}$ concentration of 0.1. Samples of 4 mL of WT at exponential phase and 8 ml of Δ*mftG* after 60 h of incubation were retrieved and further processed for RNA extraction following the manufacturer's instructions (InnuPREP RNA Mini kit 2.0 – Analytik Jena). An additional DNase step was included to remove traces of gDNA. RNA samples were then subjected to RNA-Seq library preparation following the Stranded Total RNA Prep, Ligation with Ribo-Zero Plus kit (Illumina). After depletion of the rRNA by hybridization with directed probes, the RNA samples were normalized to 500 ng/sample. The RNA samples were then subjected to fragmentation and denaturation, followed by reverse transcription through first-strand (and subsequently second-strand) cDNA synthesis. After 3′end adenylation, anchors and indexes were ligated. Barcoding was accomplished using the IDT for Illumina RNA UD Indexes Set A, Ligation (Ref# 20040553, Illumina, Berlin, Germany). Standard AMPURE XP Beads protocols (Ref# A63881; Beckman, Krefeld, Germany) were employed to purify the final library fragments, and their size and molarity were verified using a Tape Station 2200 (Agilent, Waldbronn, Germany). The equimolarly pooled libraries were subsequently subjected to a 100-cycle run on a NovaSeq 6000 apparatus (Illumina, Berlin, Germany) using an SP flowcell v. 1.5 (Ref# 20028401; Agilent, Waldbronn, Germany).

Raw reads were checked for quality using fastP (*Chen et al., 2018*), and polyX tails (-x 10), low quality ends (−3,–5, -M 25) were removed, and resulting sequences shorter than 25 were discarded (-l 25). The remaining reads were mapped to the reference genome (RefSeq accession GCF_000015005.1) using bwa-mem2 (*Vasimuddin et al., 2019*) with default parameters for the affine gap scoring model. For each sample and every gene annotation, all overlapping reads were counted and a counts table was generated for every experiment condition using custom scripts. The counts' tables were used as an input to the R package DESeq2 (*Love et al., 2014*) to compute fold changes and false discovery rates of differentially expressed genes between the control condition and each experiment condition.

## Statistical analysis

The data retrieved in this study were further plotted and analyzed statistically using GraphPad Prism 9. Significance was calculated using either two-way ANOVA with Tukey's multiple comparisons test

once interaction index was not significant, or one-way ANOVA for multiple comparisons. Significance values were plotted directly on the graphs and only more relevant comparisons are shown.

## Materials availability statement

Materials are available from the authors upon request.

## Acknowledgements

We thank Bettina Bardl (Bio Pilot Plant, Leibniz-HKI) for HPLC analysis of metabolites. We thank the Microverse Imaging Center for providing microscope facility support for data acquisition. The ELYRA 7 was funded by the Free State of Thuringia with grant number 2019 FGI 0003. The Microverse Imaging Center is funded by the Deutsche Forschungsgemeinschaft (DFG, German Research Foundation) under Germany´s Excellence Strategy - EXC 2051 - Project-ID 390713860. DAAD, German Academic Exchange Service funding is greatly acknowledged (graduate fellowship to W K A-J, fellowship for university academics and scientists to VN). APG and GL thank the Carl Zeiss Foundation for financial support.

## Additional information

### Funding

| Funder | Grant reference number | Author |
| --- | --- | --- |
| Carl-Zeiss-Stiftung | | Ana Patrícia Graça<br>Gerald Lackner |
| Deutscher Akademischer Austauschdienst | | Walid K Al-Jammal<br>Vadim Nikitushkin |

The funders had no role in study design, data collection and interpretation, or the decision to submit the work for publication.

### Author contributions

Ana Patrícia Graça, Vadim Nikitushkin, Cláudia Vilhena, Conceptualization, Data curation, Formal analysis, Validation, Investigation, Visualization, Methodology, Writing – original draft, Writing – review and editing; Mark Ellerhorst, Data curation, Formal analysis, Validation, Investigation, Methodology, Writing – original draft, Writing – review and editing; Tilman E Klassert, Conceptualization, Resources, Formal analysis, Methodology, Writing – review and editing; Andreas Starick, Data curation, Validation, Methodology, Writing – review and editing; Malte Siemers, Data curation, Formal analysis, Validation, Methodology, Writing – review and editing; Walid K Al-Jammal, Validation, Methodology; Ivan Vilotijevic, Conceptualization, Supervision, Funding acquisition, Validation, Methodology; Hortense Slevogt, Conceptualization, Funding acquisition, Validation; Kai Papenfort, Supervision, Funding acquisition, Validation, Writing – review and editing; Gerald Lackner, Conceptualization, Resources, Supervision, Funding acquisition, Validation, Visualization, Writing – original draft, Project administration, Writing – review and editing

### Author ORCIDs

Kai Papenfort ⓘ https://orcid.org/0000-0002-5560-9804
Gerald Lackner ⓘ https://orcid.org/0000-0002-0307-8319

Reviewer #1 (Public review): https://doi.org/10.7554/eLife.97559.4.sa1
Reviewer #3 (Public review): https://doi.org/10.7554/eLife.97559.4.sa2
Reviewer #4 (Public review): https://doi.org/10.7554/eLife.97559.4.sa3
Author response https://doi.org/10.7554/eLife.97559.4.sa4

## Additional files

### Supplementary files

Supplementary file 1. Sequences of the plasmids generated in this study.

Supplementary file 2. Set of organisms encoding MftG or MftC homologs (co-occurrence table). The co-occurrence table (sheet 1) contains accession numbers of MftG and MftC candidate proteins, as well as further GMC proteins found in each organism. The phylogenetic analysis was based on all GMC enzymes including MftG listed in the co-occurrence table as well as the GMC proteins listed, under 'Reference GMC enzymes' (sheet 2).

Supplementary file 3. Gene expression analysis of *M. smegmatis* MC$^2$ 155 Δ*mftG* compared to WT grown on either ethanol or glucose as a carbon source. Sheet 1: Gene expression in Δ*mftG* mutants compared to WT of *M. smegmatis* MC2 155 grown on 10 g L$^{-1}$ ethanol as the sole carbon source. Sheet 2: Gene expression in Δ*mftG* mutants compared to WT of *M. smegmatis* MC2 155 grown on 10 g L$^{-1}$ glcuose as the sole carbon source. Black: all the genes with adjusted *P*-value ≤0.05. Grey: all the genes that with adjusted *P*-value >0.05.

MDAR checklist

### Data availability

The transcriptomic datasets produced in this study are available via the Gene Expression Omnibus (GEO) database (GEO accession numbers: GSE250373, GSM7978029 - GSM7978040).

The following dataset was generated:

| Author(s) | Year | Dataset title | Dataset URL | Database and Identifier |
|---|---|---|---|---|
| Graca AP, Klassert T, Siemers M, Lackner G | 2024 | MftG is crucial for alcohol metabolism of mycobacteria by linking mycofactocin oxidation to respiration | https://www.ncbi.nlm.nih.gov/geo/query/acc.cgi?acc=GSE250373 | NCBI Gene Expression Omnibus, GSE250373 |

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

# Appendix 1

## Appendix 1—key resources table

| Reagent type (species) or resource | Designation | Source or reference | Identifiers | Additional information |
|---|---|---|---|---|
| Strain, strain background (*Escherichia coli*) | NiCo21 (DE3) | New England BioLabs | C2529H | Derivative of BL21 (DE3) |
| Strain, strain background (Mycolicibacterium smegmatis) | MC$^2$ 155 | ATCC | 700084 | Also referred to as wild type (WT) in this paper |
| Strain, strain background (*Escherichia coli*) | TOP10 | Thermo Fischer Scientific | C404010 | Derivative of K12, cloning strain |
| Gene (Mycolicibacterium smegmatis MC$^2$ 155) | mftG | GenBank | MSMEG_1428 | Corresponding Protein_ID: WP_014877070 |
| Genetic reagent (Mycolicibacterium smegmatis MC$^2$ 155) | ΔmftG | This paper | | Deletion mutant of mftG |
| Genetic reagent (Mycolicibacterium smegmatis MC$^2$ 155) | ΔmftG-mftG | This paper | | Complement mutant of mftG |
| Genetic reagent (Mycolicibacterium smegmatis MC$^2$ 155) | WT-mftG | This paper | | Overexpression mutant of mftG |
| Genetic reagent (Mycolicibacterium smegmatis MC$^2$ 155) | ΔmftG-mftGHis$_6$ | This paper | | Overexpression mutant of His-tagged mftG in ΔmftG |
| Genetic reagent (Mycolicibacterium smegmatis MC$^2$ 155) | WT- mftABCDEF | This paper | | Overexpression mutant of mftABCDEF |
| Recombinant DNA reagent | pML2424 (plasmid vector) | *Ofer et al., 2012* | | vector for double crossover event with tdTomato, gfp-hyg cassette, and PAL5000ts |
| Recombinant DNA reagent | pML2714 (plasmid vector) | *Ofer et al., 2012* | | vector with kanamycin resistance for Cre recombinase expression and gfp-hyg cassette removal |
| Recombinant DNA reagent | pPG17 (plasmid) | This paper | | pML2424 with up and downstream regions of mftG |
| Recombinant DNA reagent | pPG20 (plasmid) | *Peña-Ortiz et al., 2020a* | | pMCpAINT derivate with kanamycin resistance, potential mycofactocin promotor, and mftF |
| Recombinant DNA reagent | pPG23 (plasmid) | This paper | | pMCpAINT derivate with kanamycin resistance, potential mycofactocin promotor, and mftABCDEF |
| Recombinant DNA reagent | pPG29 (plasmid) | This paper | | pPG20 with mftF replaced with mftG |
| Recombinant DNA reagent | pPG32 (plasmid) | This paper | | pPG29 with mftG replaced with mftGHis$_6$ |
| Recombinant DNA reagent | pPG36 (plasmid) | This paper | | pMAL-C4X with malE fused with mftG codon optimized for *E. coli* expression |
| Sequence-based reagent | GMC_up_F1 | This paper | PCR primers | GCTACACTAGTCGGTGTCGTATGTGCCGAG |
| Sequence-based reagent | GMC_up_R1 | This paper | PCR primers | GCTACATTTAAATTCAAAGTCGGCGGCTAACTC |

*Appendix 1 Continued*

| Reagent type (species) or resource | Designation | Source or reference | Identifiers | Additional information |
|---|---|---|---|---|
| Sequence-based reagent | GMC_dn_F1 | This paper | PCR primers | GCTACTTAATTAATCGAC GGCTCGATCATGC |
| Sequence-based reagent | GMC_dn_R1 | This paper | PCR primers | GCTACATGCATGTTGTC GAGGCTCCGGTG |
| Peptide, recombinant protein | MftGHis$_6$ | This paper | | MftG (WP_014877070) with C-terminal hexahistidine tag |
| Commercial assay or kit | NAD$^+$/NADH Assay Kit | Merck | Sigma-Aldrich: MAK460 | |
| Commercial assay or kit | ADP/ATP Ratio Assay | Merck | Sigma-Aldrich: MAK135 | |
| Commercial assay or kit | Acetaldehyde Assay Kit | Merck | Sigma-Aldrich: MAK321 | |
| Commercial assay or kit | InnuPREP RNA Mini kit 2.0 | Analytik Jena | Analytik Jena: 845-KS-2040010 | |
| Commercial assay or kit | Illumina Stranded Total RNA Prep, Ligation with Ribo-Zero Plus | Illumina | Illumina: 20040525 | |
| Commercial assay or kit | IDT for Illumina RNA UD Indexes Set A, Ligation | Ilumina | 20040553 | |
| Commercial assay or kit | AMPURE XP Beads | Beckman | A63881 | |
| Commercial assay or kit | Roti-Nanoquant | Carl Roth | Carl Roth: K880 | Protein Concentration Determination Kit |
| Chemical compound, drug | premycofactocinone (PMFT) | *Ellerhorst et al., 2022* | | |
| Chemical compound, drug | premycofactocinol (PMFTH$_2$) | *Ellerhorst et al., 2022* | | |
| Chemical compound, drug | methylmycofactocinol-2 (MMFT-2H$_2$) | This study | | methylmycofactocinol-2 purified from WT- mftABCDEF |
| Chemical compound, drug | cellulase from Trichoderma reesei ATCC 26921 | Sigma-Aldrich | Sigma-Aldrich: C8546 | |
| Chemical compound | HADA | Bio-Techne | Bio-Techne: 6647 | 3-[[(7-Hydroxy-2-oxo-2H-1-benzopyran-3-yl)carbonyl]amino] -D-alanine hydrocholoride |
| Chemical compound | NADA | Bio-Techne | Bio-Techne: 6648 | 3-[(7-Nitro-2,1,3-benzoxadiazol-4-yl)amino] -D-alanine hydrochloride |
| Chemical compound | RADA | Bio-Techne | Bio-Techne: 6649 | (S)-N-(9-(4-((2-amino-2-carboxyethyl)carbamoyl)-2-carboxyphenyl) -6-(dimethylamino)3 H-xanthen-3-ylidene)-N-methylmethanaminium |
| Chemical compound, drug | Tyloxapol | BioXtra (Sigma-Aldrich) | Sigma-Aldrich: T0307 | |
| Chemical compound, drug | Tween 80 | Sigma-Aldrich | Sigma-Aldrich: P1754 | |
| Chemical compound, drug | 3,3'-diethyloxacarbocy-anine iodide (DIOC$_2$(3)) | Sigma-Aldrich | Sigma-Aldrich: 320684 | |
| Chemical compound, drug | Carbonyl cyanide 3-chlorophenylhydrazone (CCCP) | Sigma-Aldrich | Sigma-Aldrich: C2759 | |
| Chemical compound, drug | Isopropyl-β -D-thiogalacto-pyranoside (IPTG) | Carl Roth | Carl Roth: 2316.3 | |
| Chemical compound, drug | Flavine adenine dinucleotide disodium salt (FAD) | Carl Roth | Carl Roth: 5581.1 | |
| Chemical compound, drug | β-Nicotiamid adenin dinucleotide (NAD) hydrate | Sigma-Aldrich | Sigma-Aldrich: N1511 | |
| Chemical compound, drug | β-Nicotiamid adenin dinucleotide (NADH) disodium salt | Carl Roth | Carl Roth: AE12 | |
| Chemical compound, drug | N,N-Dimethyl-4-nitrosoaniline (NDMA) | Sigma-Aldrich | Sigma-Aldrich: D172405 | Electron Acceptor |

*Appendix 1 Continued*

| Reagent type (species) or resource | Designation | Source or reference | Identifiers | Additional information |
|---|---|---|---|---|
| Chemical compound, drug | 2,6-Dichloro-phenolindophenol sodium salt hydrate | Sigma-Aldrich | Sigma-Aldrich: D1878 | Electron Acceptor |
| Chemical compound, drug | Phenazine methosulfate | Sigma-Aldrich | Sigma-Aldrich: P7625 | Electron Acceptor |
| Chemical compound, drug | Potassium cyanide (KCN) | Sigma-Aldrich | Sigma-Aldrich: 60178 | |
| Other, FPLC column | MBPTrap HP 1 mL | Cytiva | Cytiva: 29048641 | |
| Other, FPLC column | Superdex 30 Increase 10/300 GL | Cytiva | Cytiva: 29219757 | |
| Other, HPLC column | Kinetex 2.6 µm XB-C18 100 Å LC-Column, 150x2.1 mm | Phenomenex | Phenomenex: 00F-4496-AN | Column for LC-MS/MS |
| Other, HPLC column | SecurityGuard ULTRA Cartridge, UHPLC C18, 2.1 mm | Phenomenex | Phenomenex: AJ0-8782 | Guard Column for LC-MS/MS Column |
| Other, HPLC column | Kromasil 5 µm C18 100 Å LC-Column, 40x4 mm | Dr. Maisch GmbH | Dr. Maisch GmbH: k15.9e.s0404 | Guard Column for HPLC-RI/-UV Column |
| Other, HPLC column | Aminex HPX-87H 9 µm Ion Exclusion Column, 300x7.8 mm, 9 µm | Bio-Rad | Bio-Rad: #1250140 | Column for HPLC-RI/-UV |
| Other, solid phase extraction (SPE) column | CHROMABOND C18, 45 µm, 70 mL/10,000 mg | Machery-Nagel | Machery-Nagel: 730261 | |
| Software, algorithm | ggVennDiagram 1.2.3 | *Aleksenko et al., 2020* | | R Package for Venn Diagram Construction |
| Software, algorithm | FastTree 2.1.11 | *Price et al., 2010* | | Software for fast phylogenetic analysis using a Maximum Likelihood algorithm |
| Software, algorithm | Geneious Prime 2022.2.2 | Dotmatics | RRID:SCR_010519 | Molecular Biology software |
| Software, algorithm | GraphPad Prism 9 | Dotmatics | RRID:SCR_002798 | Statistics software |
| Software, algorithm | CGQuant | Aquila Biolabs | | Software for processing data acquired by CGQ (Cell Growth Quantifier) instrument |
| Software, algorithm | mzMine 2.53 | *Pluskal et al., 2010* | | Metabolomics software |
| Software, algorithm | fastP | *Chen et al., 2018* | | |
| Software, algorithm | BWA-MEM | *Vasimuddin et al., 2019* | | |
| Software, algorithm | DESeq2 | *Love et al., 2014* | | |
| Software, algorithm | flowCore_2.2.0 | *Hahne et al., 2009* | | Analysis of flow-cytometric data |
| Software, algorithm | ggcyto_1.18.0 | *Van et al., 2018* | | Analysis of flow-cytometric data |
| Software, algorithm | ChimeraX 1.2.5 | *Goddard et al., 2018* | | Protein visualization and structure analysis software |
| Software, algorithm | BLAST | *Altschul et al., 1990* | | Biological sequence similarity search software |
| Software, algorithm | FlowJo v10.8 | BD Life Sciences | | Flow cytometry software |
| Software, algorithm | enviPat | *Loos et al., 2015* | | |
| Software, algorithm | ZenBlack | Zeiss | | Software for image analysis and acquisition |
| Software, algorithm | Fiji ImageJ | *Schindelin et al., 2012* | | Software for image analysis |
| Software, algorithm | FlowJo v10.8 | BD Life Sciences | | Software for flow cytometry data acquisition and analysis |
| Software, algorithm | Oxytherm+ | Hansatech Instruments | | Respirometer software |

