## [Editor Report · eLife Assessment]

Graca et al. reports a **fundamental** missing link in the ethanol metabolism of mycobacteria and illuminates the role of a flavoprotein dehydrogenase that acts as an electron shuttle between an uncommon redox cofactor and the electron transport chain. Overall, the data presented are **compelling**, supported by a range of well designed and meticulous experiments. The findings will be of broad interest to researchers investigating bacterial metabolism.

---

## [Referee Report · Reviewer #1 (Public review)]

Using genetically engineered Mycolicibacterium smegmatis strains, the authors tried to decipher the role of the last gene in the mycofactocin operon, mftG. They found that MftG was essential for growth in the presence of ethanol as the sole carbon source, but not for the metabolism of ethanol, evidenced by the equal production of acetaldehyde in the mutant and wild type strains when grown with ethanol (Fig 3). The phenotypic characterization of ΔmftG cells revealed a growth-arrest phenotype in ethanol, reminiscent of starvation conditions (Fig 4). Investigation of cofactor metabolism revealed that MftG was not required to maintain redox balance via NADH/NAD^+^, but was important for energy production (ATP) in ethanol. Since mycobacteria cannot grow via substrate-level phosphorylation alone, this pointed to a role of MftG in respiration during ethanol metabolism. The accumulation of reduced mycofactocin points to impaired cofactor cycling in the absence of MftG, which would impact the availability of reducing equivalents to feed into the electron transport chain for respiration (Fig 5). This was confirmed when looking at oxygen consumption in membrane preparations from the mutant and wild type strains with reduced mycofactocin electron donors (Fig 7). The transcriptional analysis supported the starvation phenotype, as well as perturbations in energy metabolism.

The link between mycofactocin oxidation and respiration is shown by whole-cell and membrane respiration measurements. I look forward to seeing what the electron acceptor/s are for MftG. Overall, the data and conclusions support the role of MftG in ethanol metabolism as a mycofactocin redox enzyme.

---

## [Referee Report · Reviewer #3 (Public review)]

Summary:

The work by Graca et al. describes a GMC flavoprotein dehydrogenase (MftG) in the ethanol metabolism of mycobacteria and provides evidence that it shuttles electrons from the mycofactocin redox cofactor to the electron transport chain.

Strengths:

Overall, this study is compelling, exceptionally well-designed and thoroughly conducted. An impressively diverse set of different experimental approaches is combined to pin down the role of this enzyme and scrutinize the effects of its presence or absence in mycobacteria cells growing on ethanol and other substrates. Other strengths of this work are the clear writing style and stellar data presentation in the figures, which makes it easy also for non-experts to follow the logic of the paper. Overall, this work therefore closes an important gap in our understanding of ethanol oxidation in mycobacteria, with possible implications for the future treatment of bacterial infections.

Weaknesses:

I see no major weaknesses in this work, which in my opinion leaves no doubt about the role of MftG.

---

## [Referee Report · Reviewer #4 (Public review)]

Summary:

The manuscript by Graça et al. explores the role of MftG in the ethanol metabolism of mycobacteria. The authors hypothesise that MftG functions as a mycofactocin dehydrogenase, regenerating mycofactocin by shuttling electrons to the respiratory chain of mycobacteria. Although the study primarily uses *M. smegmatis* as a model microorganism, the findings have more general implications for understanding mycobacterial metabolism. Identifying the specific partner to which MftG transfers its electrons within the respiratory chain of mycobacteria would be an important next step, as pointed out by the authors.

Strengths

The authors have used a wide range of tools to support their hypothesis, including co-occurrence analyses, gene knockout and complementation experiments, as well as biochemical assays and transcriptomics studies.

An interesting observation that the mftG deletion mutant grown on ethanol as the sole carbon source exhibited a growth defect resembling a starvation phenotype.

MftG was shown to catalyse the electron transfer from mycofactocinol to components of the respiratory chain, highlighting the flexibility and complexity of mycobacterial redox metabolism.

The authors have taken on the majority of recommendations by the reviewers and made changes in the manuscript accordingly. I don't have any further suggestions.

---

## [Author Response]

The following is the authors’ response to the previous reviews.

**Public Reviews:**

**Reviewer #1 (Public review):**
Using a knock-out mutant strain, the authors tried to decipher the role of the last gene in the mycofactocin operon, mftG. They found that MftG was essential for growth in the presence of ethanol as the sole carbon source, but not for the metabolism of ethanol, evidenced by the equal production of acetaldehyde in the mutant and wild type strains when grown with ethanol (Fig 3). The phenotypic characterization of ΔmftG cells revealed a growth-arrest phenotype in ethanol, reminiscent of starvation conditions (Fig 4). Investigation of cofactor metabolism revealed that MftG was not required to maintain redox balance via NADH/NAD^+^, but was important for energy production (ATP) in ethanol. Since mycobacteria cannot grow via substrate-level phosphorylation alone, this pointed to a role of MftG in respiration during ethanol metabolism. The accumulation of reduced mycofactocin points to impaired cofactor cycling in the absence of MftG, which would impact the availability of reducing equivalents to feed into the electron transport chain for respiration (Fig 5). This was confirmed when looking at oxygen consumption in membrane preparations from the mutant and would type strains with reduced mycofactocin electron donors (Fig 7). The transcriptional analysis supported the starvation phenotype, as well as perturbations in energy metabolism, and may be beneficial if described prior to respiratory activity data.The data and conclusions support the role of MftG in ethanol metabolism.

We thank the reviewer for the positive evaluation of our manuscript.

**Reviewer #3 (Public review):**
Summary:The work by Graca et al. describes a GMC flavoprotein dehydrogenase (MftG) in the ethanol metabolism of mycobacteria and provides evidence that it shuttles electrons from the mycofactocin redox cofactor to the electron transport chain.Strengths:Overall, this study is compelling, exceptionally well designed and thoroughly conducted. An impressively diverse set of different experimental approaches is combined to pin down the role of this enzyme and scrutinize the effects of its presence or absence in mycobacteria cells growing on ethanol and other substrates. Other strengths of this work are the clear writing style and stellar data presentation in the figures, which makes it easy also for non-experts to follow the logic of the paper. Overall, this work therefore closes an important gap in our understanding of ethanol oxidation in mycobacteria, with possible implications for the future treatment of bacterial infections.Weaknesses:I see no major weaknesses of this work, which in my opinion leaves no doubt about the role of MftG.

We thank the reviewer for the positive evaluation of our manuscript.

**Reviewer #4 (Public review):**
Summary:The manuscript by Graça et al. explores the role of MftG in the ethanol metabolism of mycobacteria. The authors hypothesise that MftG functions as a mycofactocin dehydrogenase, regenerating mycofactocin by shuttling electrons to the respiratory chain of mycobacteria. Although the study primarily uses *M. smegmatis* as a model microorganism, the findings have more general implications for understanding mycobacterial metabolism. Identifying the specific partner to which MftG transfers its electrons within the respiratory chain of mycobacteria would be an important next step, as pointed out by the authors.Strengths:The authors have used a wide range of tools to support their hypothesis, including co-occurrence analyses, gene knockout and complementation experiments, as well as biochemical assays and transcriptomics studies.An interesting observation that the mftG deletion mutant grown on ethanol as the sole carbon source exhibited a growth defect resembling a starvation phenotype.MftG was shown to catalyse the electron transfer from mycofactocinol to components of the respiratory chain, highlighting the flexibility and complexity of mycobacterial redox metabolism.Weaknesses:Could the authors elaborate more on the differences between the WT strains in Fig. 3C and 3E? in Fig. 3C, the ethanol concentration for the WT strain is similar to that of WT-mftG and ∆mftG-mftG, whereas the acetate concentration in thw WT strain differs significantly from the other two strains. How this observation relates to ethanol oxidation, as indicated on page 12.

This is a good question, and we agree with the reviewer that the sum of processes leading to the experimental observations shown in Figure 3 are not completely understood. For instance, when looking at ethanol concentrations, evaporation is a dominating effect and the situation is furthermore confounded by the fact that the rate of ethanol evaporation appears to be inversely correlated to the optical density of the samples (see Figure 3E and compare media control as well as the samples of D*mftG* and D*mftG* at OD_600_ = 1). Additionally, the growth rate and thus the OD_600_ of all strains monitored are different at each time point, thus further complicating the analysis. This is why we assume that the rate of ethanol oxidation is mirrored more clearly by acetate formation, at least in the early phase before 48 h (Figure 3E),i.e., before acetate consumption becomes dominant in D*mftG-mftG* and WT-*mftG*. Here, we see that the rate of acetate formation is zero for media controls, low for D*mftG,* but high for WT as well as D*mftG-mftG* and WT-*mftG.* The latter two strains also showed an earlier starting point of growth as well as acetate formation and the following phase of acetate depletion.

All of these observations are in line with our general statement, i.d., “Parallel to the accelerated and enhanced growth described above (Figure 3A), the overexpression strains displayed higher rates of ethanol consumption as well as an earlier onset of acetate overflow metabolism and acetate consumption (Figure 3D).” We are still convinced that this summary describes the findings well and avoids unnecessary speculation.

The authors conclude from their functional assays that MftG catalyses single-turnover reactions, likely using FAD present in the active site as an electron acceptor. While this is plausible, the current experimental set up doesn't fully support this conclusions, and the language around this claim should be softened.

This is a fair point. We revised our claim accordingly. In particular, we changed:

Page 28: we added “possibly”

Page 28 we changed “single-turnover reactions” to “reactions reminiscent of a single-turnover process”.

The authors suggest in the manuscript that the quinone pool (page 24) may act as the electron acceptor from mycofactocinol, but later in the discussion section (page 30) they propose cytochromes as the potential recipients. If the authors consider both possibilities valid, I suggest discussing both options in the manuscript.

This is true. However, no change to the manuscript is necessary, since both options were discussed on page 30.

**Recommendations for the authors:**

**Reviewer #1 (Recommendations for the authors):**
The authors addressing some of the original recommendations is appreciated e.g. title change. Other recommendations that were not adequately addressed would mostly improve the clarity and help comprehension for the reader, but they are at the author's discretion.
**Reviewer #3 (Recommendations for the authors):**
Abstract: "Here, we show that MftG enzymes strictly require mft biosynthetic genes and are found in 75% of organisms harboring these genes". I read this sentence several times and I am still somewhat confused and not sure what exactly is meant here. I suggest to rephrase, e.g., to "Here, we show that in 75% of all organisms that harbour the mft biosynthetic genes, MftG enzymes are also encoded and functionally associated with these genes" (if that was meant; also the abbreviation mft should be introduced in the abstract or otherwise the full name be used).

We thank the reviewer for the good hint. We changed the sentence to “Here, we show that MftG enzymes are almost exclusively found in genomes containing mycofactocin biosynthetic genes and are present in 75% of organisms harboring these genes”.

p.3, 2nd paragraph: "Although the role of MFT in alcohol metabolism is well established, further biological roles of mycofactocin appear to exist." Mycofactocin is once written as MFN and once in full length, which is slightly confusing. Consider rephrasing, e.g., to "...further biological roles of this cofactor appear to exist".

Thank you, we adopted the suggested change.

Fig. 1: Consider adding MftG in brackets after "mycofactocin dehydrogenase" in panel B.

Good suggestion. We added (MftG) to the figure.

Fig. 3: Legend should be corrected. The color of the signs should be teal diamond for "*M. smegmatis* double presence of the mftG gene" and orange upward facing triangle for "Medium with 10 g L-1 of ethanol without bacterial inoculation". Aside from the coloration, the order should ideally also be identical to the one shown in the upper right part.

Thank you for the valuable hint! We corrected the legend and unified the legends in the figure caption and figure.

p.20 : It is not exactly clear to me why "semipurified cell-free extracts from *M. smegmatis* ∆mftG-mftGHis6 " were used here rather than the purified enzyme. Was the purification by HisTrap columns not feasible or was the protein unstable when fully purified? In any case, it would help the reader to quickly state the reason in this section.

Indeed, the problem with *M. smegmatis* as an expression host was a combination of low protein yield and poor binding to Ni-NTA columns. In *E. coli*, poor expression, low solubility or poor binding was the issue. Unfortunately, the usage of other affinity tags resulted in either poor expression or inactive protein. We have shortly mentioned the major issues on page 21 and prefer not to focus on failed attempts too much.

p. 21: "We, therefore, concluded that MftG can indeed interact with mycofactocins as electron donors but might require complex electron acceptors, for instance, proteins present in the respiratory chain." I agree. For the future it might be worthwhile to determine the redox potential of MftG, which could provide hints on the natural electron acceptor.

Thank you for the suggestion. We will consider this question in our future work.

p. 23: "In *M. smegmatis*, cyanide is a known inhibitor of the cytochrome bc/aa3 but not of cytochrome bd (34), therefore, the decrease of oxygen consumption when MFTs were added to the membrane fractions in combination with KCN (Figure 7), revealed that MFT-induced oxygen consumption is indeed linked to mycobacterial respiration." It might be a good idea to quickly recapitulate the functions of these cytochromes here. Also, I think it should read "bc1aa3" (also correct in legend of Fig. 8 that says "bcc-aa3").

Thank you for the good observation. We changed all instances to the correct designation (bc1-aa3).

**Reviewer #4 (Recommendations for the authors):**
Abstract: revise the wording "MftG enzymes strictly require mft biosynthetic genes". It should be either mftG gene with the mft biosynthetic genes or MftG enzyme with the Mft biosynthetic proteins. I also suggest replacing "require" with a more appropriate term.

This was taken care of. See above.

Page 3, end of the first paragraph; does the alcohol dehydrogenase refer to Mno/Mdo?

Partially, yes, but also to other alcohol dehydrogenases.

Page 4, radical SAM; define upon first use

Good, point, we changed “radical SAM” to radical *S*-adenosyl methionine (rSAM)

Page 6; Rossman fold refers to the fold and not only the FAD binding pocket.

Good point. We deleted “(Rossman fold)”

Page 11; not exactly sure what this means "the growth curve of the complemented strain, which could be dysregulated in mftG expression"

By “dysregulated” expression, we mean that the expression of *mftG* could be higher or lower than in the WT and could follow different regulatory signals than in the wild type. Since this phenomenon is not well understood, we would like to avoid speculative discussions.

Page 11; Figures 2E and 2C should be 3E and 3C. Likewise on page 12 Figure 2D.

Thank you very much for the valuable hint. We corrected the figure numbers as suggested.

Page 12; the last Figure 3D in the page should be 3E?

Yes, good catch, we corrected the Figure number.

Page 17, KO; define upon first use.

Good suggestion, we changed both instances of “KO” to “knockout”

Page 24; revise: "for instance. For example"

We deleted “for instance”.

Page 26; change 6.506 to 6,506

Corrected.

Page 23; "In *M. smegmatis*, cyanide is a known inhibitor ..." is too long and not easy to understand/follow.

Good suggestion. We simplified the sentence to “Therefore, the decrease of oxygen consumption in the presence of KCN (Figure 7) revealed…”

Page 29; "single-turnover reactions could be observed". There are no experiments to support this statement, except the results shown in Figure 7F. I suggest softening the language, as it has been done on page 21. To claim single-turnover, a proper kinetic analysis would be necessary, which is not included in the current manuscript.

This is true and has been taken care of. See above.

Figure 1; Indicate mycofactocin dehydrogenase as MftG

Done.

Figure 5A; what is the significance of comparing ∆mftG glucose with WT ethanol?

We agree, that, although the difference of the two columns is significant, this does not have any relevant meaning. Therefore, we removed the bracket with p-value in Panel A.

Make HdB-Tyl/HdB-tyloxapol usage consistent throughout the document. Likewise, re the usage of mycobacteria/Mycobacteria/Mycobacteria

Thank you for the valuable hint, we unified the usage throughout the document